# RDumb: A simple approach that questions our progress in continual test-time adaptation

**Ori Press**[1][*] **Steffen Schneider**[1,2] **Matthias Kümmerer**[1][†] **Matthias Bethge**[1][†]

[1]University of Tübingen, Tübingen AI Center, Germany
[2]EPFL, Geneva, Switzerland

## Abstract

Test-Time Adaptation (TTA) allows to update pre-trained models to changing data distributions at deployment time. While early work tested these algorithms for individual fixed distribution shifts, recent work proposed and applied methods for continual adaptation over long timescales. To examine the reported progress in the field, we propose the Continually Changing Corruptions (CCC) benchmark to measure asymptotic performance of TTA techniques. We find that eventually all but one state-of-the-art methods collapse and perform worse than a non-adapting model, including models specifically proposed to be robust to performance collapse. In addition, we introduce a simple baseline, "RDumb", that periodically resets the model to its pretrained state. RDumb performs better or on par with the previously proposed state-of-the-art in all considered benchmarks. Our results show that previous TTA approaches are neither effective at regularizing adaptation to avoid collapse nor able to outperform a simplistic resetting strategy.

## 1 Introduction

Biological vision is remarkably robust at adapting to continually changing environments. Imagine cycling through the forest on a cloudy day and observing the world around you: You will encounter a wide variety of animals and objects, and be able to recognize them without effort. Even as the weather changes, rain sets in, or you start cycling faster, the human visual system effortlessly adapts and robustly estimates the surroundings [45]. Equipping machine vision with similar capabilities is a long-standing and unsolved challenge, with numerous applications in autonomous driving, medical imaging, and quality control, to name a few.

Techniques for improving the robustness to domain shifts of ImageNet-scale [37] classification models include pre-training of large models on diverse and/or large-scale datasets [26, 33, 50] and robustification of smaller models by specifically designed data augmentation [11, 14, 36]. While these techniques are applied during training time, recent work [8, 28, 29, 30, 35, 38, 47, 48] explored possibilities of further adapting models by Test-Time Adaptation (TTA). Such methods continuously update a given pretrained model exclusively using their input data, without access to its labels. Test-time entropy minimization (Tent; 47) has become a foundation for state-of-the-art TTA methods. Given an input stream of images, Tent updates a pretrained classification model by minimizing the entropy of its outputs, thereby continuously increasing the model's confidence in its predictions for every input image.

Previous TTA work [8, 28, 29, 35, 38, 47, 54] evaluate their models on ImageNet-C [13] or smaller scale image classification benchmarks [18, 20]. ImageNet-C consists of 75 copies of the ImageNet validation set, wherein each copy is corrupted according to 15 different noises at 5 different severity

---

[*]ori.press@bethgelab.org. [†]Joint senior authors. Code: https://github.com/oripress/CCC.

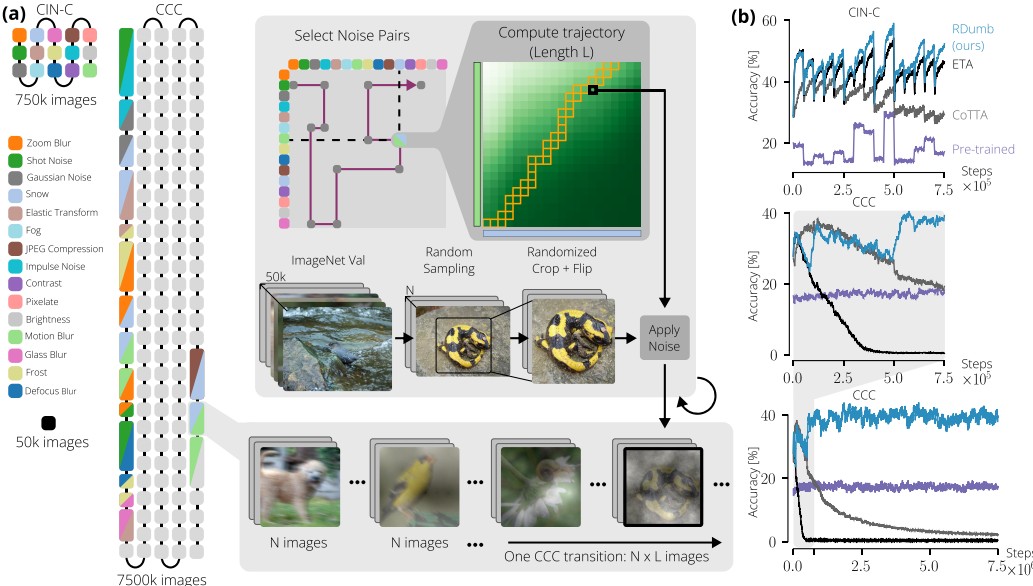

Figure 1: Continuously Changing Corruptions show limitations of existing TTA methods. (a) Comparison between ImageNet-Val, CIN-C and CCC. The proposed version of CCC is $10\times$ longer than CIN-C and could naturally be extended even further without repeating images. CCC consists of sequences of smooth transitions from one ImageNet-C noise to another one. For each such pair, we construct a trajectory continuously interpolating from one pure noise to the other pure noise such that baseline accuracy is kept constant. For each point along the trajectory, we sample a batch of 1k, 2k, or 5k images from ImageNet-Val, randomly crop and flip it and apply the noise combination. (b) Due to its short length and high variability in difficulty, CIN-C (top) is unable to reveal the collapse of methods such as ETA and CoTTA, while CCC (middle and bottom) can.

levels. When TTA models are evaluated on ImageNet-C, they are adapted on each noise and severity combination individually starting from their pretrained weights. Such a one-time adaptation approach is of little relevance when it comes to deploying TTA models in realistic scenarios. Instead, stable performance over a long run time after deployment is the desirable goal.

TTA methods are by design readily applicable to this setting and recently the field has started to move towards testing TTA models in continual adaptation settings [7, 30, 48]. Strikingly, this revealed that the dominant TTA approach Tent [47] decreases in accuracy over time, eventually being less accurate than a non-adapting, pretrained model [30, 48]. In this work, we refer to any model whose classification accuracy falls below that of a non-adapting, pretrained model, as having "collapsed".

This collapsing behaviour of Tent shows that it cannot be used in continual adaptation over long time scales without modifications. While previous benchmarking of TTA methods already managed to reveal the collapse of Tent, our work shows that in fact all current TTA methods collapse sooner or later, *including methods with explicit built-in anti-collapse strategies*.

Since current benchmarks have not been sufficient to detect collapse in several models, we introduce an image classification benchmark designed to thoroughly evaluate TTA models for their long-term behavior. Our benchmark, *Continuously Changing Corruptions* (CCC), tests models on their ability to adapt to image corruptions that are constantly changing, much like when fog turns to rain or day turns to night. CCC allows us to easily control different factors that could affect the ability of a given method to continuously adapt: the corruptions and their order, the difficulty of the images themselves, and the speed at which corruptions transition. Most importantly, the length of our benchmark is ten times longer than that of previous benchmarks, and more diverse by including all kinds of combinations of corruptions (see Figure 1a). Using CCC, we discover that seven recently published state-of-the-art TTA methods are less accurate than a non-adapting, pretrained model. While Tent was already shown to collapse [7, 30, 48], we show that this problem is not specific to

Tent, and that many other methods – including specifically designed continual adaptation methods – collapse as well.

Finally, we propose "*RDumb*" [1] as a minimalist baseline mechanism that simply *Resets* the model to its pretrained weights at regular intervals. Previous work employs more sophisticated methods combining entropy minimization with various regularization approaches, yet we show that RDumb is superior on both existing benchmarks and ours (CCC). Our results call the progress made in continual TTA so far into question, and provide a richer set of benchmarks for realistic evaluation of future methods.

Our contributions are:

- We introduce the continual adaptation benchmark CCC. We show that previous benchmarks are too short to meaningfully assess long-term continual adaptation behaviour, and are too uncontrolled to assess the short-term learning dynamics.

- Using CCC, we show that the performances of all but one current TTA methods drop below a non-adapting, pre-trained baseline when trained over long timescales.

- We propose "*RDumb*" as a baseline and show that it outperforms all previous methods with a minimalist resetting strategy.

## 2  CCC: Towards Infinite Testing with Continuously Changing Corruptions

Until recently, it was common to evaluate TTA methods only on datasets on individual domain shifts such as the corruptions of ImageNet-C [13]. However, the world is steadily changing and recently the community started moving towards continual adaptation, i.e., evaluating methods with respect to their ability to adapt to ongoing domain shifts [30, 42, 48].

The dominant method of evaluating continual adaptation on ImageNet scale is to concatenate the top severity datasets of the 15 ImageNet-C corruptions into one big dataset. We refer to the variant of this dataset introduced by [48] as *Concatenated ImageNet-C* (CIN-C). CIN-C was used to demonstrate the collapse of Tent and the stability of recent TTA methods by [7, 30, 48].

In Figure 1b, we evaluate a range of TTA methods on CIN-C and notice three potential problems: Firstly, ETA [30] appears to be stable and better than a non-adapting, pretrained baseline, but is revealed to collapse when tested on CCC. Additionally, while CoTTA[48] clearly goes down in performance, it is not yet clear whether it collapses or stabilizes above or below baseline performance. Fundamentally, CIN-C turns out to be too short to yield reliable, conclusive results. Secondly, assessing adaptation dynamics is further obscured by the considerable variations of the baseline performance among the different corruptions in CIN-C. This is not only a factor that affects adaptation itself (shown in [30, 47]), it also leads to substantial fluctuations in performance across multiple runs, making it difficult to obtain a clear and reliable assessment. Finally, CIN-C features exclusively abrupt transitions between different corruption types. In contrast, in the real world, domain changes may often be smooth and subtle with varying speeds: day to night, rain to sunshine, or the accumulation of dust on a camera. Therefore, it is important to also probe TTA methods on continual domain changes that are not tied to a specific point in time and thus constitute a relevant test for stable continual adaptation.

Here we propose a new benchmark, *Continuously Changing Corruptions* (CCC), to address these issues. CCC solves the issues of benchmark length, uncontrolled baseline difficulty, and transition smoothness in a simple and effective manner. Firstly, the length issue is remedied because the individual runs of CCC are constructed by a generation process which can generate very long datasets without reusing images. In this work we use runs of 7.5M images, which is 10 times as long as CIN-C. If required to compare methods in future work (where collapse is even slower), it is straightforward to generate even longer benchmarks within the CCC framework. Secondly, since both [30, 47] have shown that dataset difficulty is a confounder when studying adaptation, the difficulty of individual benchmark runs is kept stable. Additionally, we examine three different difficulty levels to ensure a comprehensive yet controlled evaluation. Finally, CCC exhibits smooth domain shifts: it applies two corruptions to each image. Over time, the severity of one corruption is smoothly increased while the

---

[1]The name was inspired by GDumb [32].

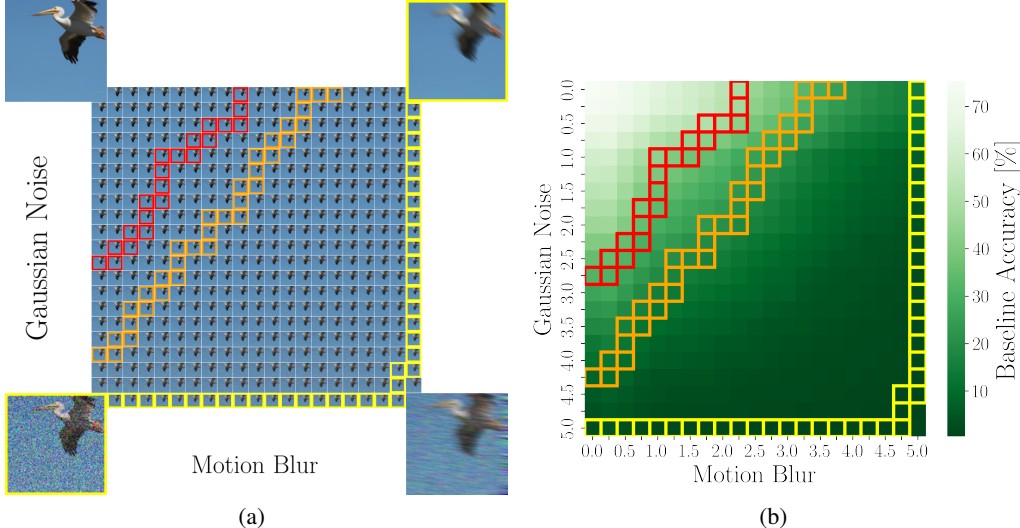

(a)                 (b)

Figure 2: (a) Each corruption of CCC consists of applying two ImageNet-C corruptions at different severities. We extend the individual severities to be more fine-grained than in ImageNet-C, allowing for smoother noise changes, and exponentially more (noise, severity) combinations. The corners are enlarged for easier viewing, zoom in for greater detail. (b) Sample dataset sequences with a constant baseline accuracy. The sequences start from the left where Motion Blur is zeroed out, and end at the top with Gaussian noise zeroed out. The colors red, orange, and yellow correspond to trajectories in CCC-Easy, CCC-Medium and CCC-Hard, respectively.

severity of the other is decreased, maintaining the desired difficulty. We also study three different speeds for applying this process. We will now outline the generation procedure of the dataset.

**Continuously changing image corruptions** To allow smooth transitions between corruptions, we introduce a more fine-grained severity level system to the ImageNet-C dataset. We interpolate the parameters of the original corruptions (integer-valued severities from 1 to 5) to finer grained severity levels from 0 to 5 in steps of 0.25. We apply two different ImageNet-C corruptions to each image, such that we can decrease the severity of one corruption while increasing the severity of another one. Hence, the corruptions of CCC are given by quadruples $(c_1, s_1, c_2, s_2)$, where $c_1$ and $c_2$ are ImageNet-C corruption types and $s_1$ and $s_2$ are severity levels. When applying such a corruption, we first apply $c_1$ and then $c_2$ at their respective severities (see Figure 2a).

**Calibration to desired baseline accuracy** In order to control baseline accuracy, we need to know how difficult each combination of 2 noises and their respective severities is. To that end, we first select a subset of 5,000 images from the ImageNet validation set. For each corruption $(c_1, s_1, c_2, s_2)$, we corrupt all 5,000 images accordingly and evaluate the resulting images with a pre-trained ResNet-50 [10]. The resulting accuracy is what we refer to as *baseline accuracy* and what we use for controlling difficulty. In total, we evaluate more than 463 million corrupted images. Previous work, [12], measures normalized accuracy using AlexNet [19], which is less pertinent in present-day contexts. In addition, the accuracy of non-adapting Vision Transformers are stable on CCC as well (Figure 4).

**Generating Benchmark Runs** Having calibrated the corruptions pairs, we prepare benchmark runs with different baseline accuracies, transition speeds, and noise orderings. We pick 3 different baseline accuracies: 34%, 17%, and 2% (CCC-Easy, CCC-Medium, CCC-Hard respectively). For each one of the difficulties, we select a further 3 transition speeds: 1k, 2k, 5k. Lastly, for each difficulty and transition speed combination we use 3 different noise orderings, determined by 3 random seeds. To generate each run, we first select the initial corruption at the severity which according to our calibration is closest to the desired baseline accuracy. We then transition to the second corruption of the noise ordering by repeatedly either decreasing the severity of the first noise by 0.25 or increasing the severity of the second noise by 0.25 such that the baseline accuracy is as close to the target as possible (see Figure 2). In each step along each path, we sample 1k, 2k, or 5k images from the

ImageNet validation set depending on the desired transition speed. Each image is randomly cropped and flipped for increasing the diversity of the dataset, and then corrupted.

Once the path from the initial to the second corruption is finished, the process is repeated for transitioning to the third corruption and so on (for more details see Appendix B). In the end, we have 3 difficulties consisting of 9 benchmark runs each. CCC-Medium at a speed of 2k corresponds roughly to CIN-C's difficulty and transition speed.

## 3 RDumb: Turning your model off and on again

Continual test-time adaptation needs to successfully adapt models over arbitrarily long timescales during deployment. Resetting a model to its initial weights at fixed intervals fulfills this criterion by design, yet allows to benefit from adaptation over short time scales. Surprisingly, such an approach has never been tried before (see Appendix E for more discussion).

Regarding the choice of the adaptation loss, we build on the weighted entropy used in ETA [30]. For a stream of input images $\mathbf{x}_1, \mathbf{x}_2, \ldots$, we compute class probabilities $\mathbf{y}_t = f_{\Theta_t}(\mathbf{x}_t)$ and optimize the loss function

$$L(\mathbf{y}_t; \overline{\mathbf{y}}_{t-1}) = \left( \frac{\mathbb{1}[(|\cos(\mathbf{y}_t, \overline{\mathbf{y}}_{t-1})| < \epsilon) \wedge (H(\mathbf{y}_t) < H_0)]}{\exp(H(\mathbf{y}_t) - H_0)} \right) H(\mathbf{y}_t) \tag{1}$$

which weights the entropy $H(\mathbf{y}_t) = -\mathbf{y}_t^\top (\log \mathbf{y}_t)$ of each prediction using the similarity to averaged previously predicted class probabilities, $\overline{\mathbf{y}}_t = (\mathbf{y}_1 + \cdots + \mathbf{y}_t)/t$, and a comparison to a fixed entropy threshold $H_0$. $\cos(\mathbf{u}, \mathbf{v})$ refers to the cosine similarity between vectors $\mathbf{u}$ and $\mathbf{v}$. At each step, (part of) the weights $\Theta_t$ are updated using the Adam optimizer [17]. At every $T$-th step, $\Theta_t$ is reset to the baseline weights $\Theta_0$. We use $\epsilon = 0.05$ and $H_0 = 0.4 \times \ln 10^3$ following [30], and select $T = 1000$ based on the holdout noises in IN-C (see Section 6).

## 4 Experiment Setup

We benchmark RDumb alongside a range of recently published TTA models. For all models, we use a batch size of 64. In all models, the BatchNorm statistics are estimated on the fly, and the affine shift and scale parameters are optimized according to a model-specific strategy outlined below.

- **BatchNorm (BN) Adaptation** [29, 38] estimates the BatchNorm statistics (mean and variance) separately for each batch at test time. The affine transformation parameters are not adapted.

- **Tent** [46] optimizes the entropy objective on the test set in order to update the scale and shift parameters of BatchNorm (in addition to learning the statistics).

- **Robust Pseudo-Labeling (RPL)** [35] uses a teacher-student approach in combination with a label noise resistant loss.

- **Conjugate Pseudo Labels (CPL)** [8] use meta learning to learn the optimal adaptation objective function across a class of possible functions.

- **Soft Likelihood Ratio (SLR)** [28] uses a loss function that is similar to entropy, but without vanishing gradients. *Anti-Collapse Mechanism:* An additional loss is used to encourage the model to have uniform predictions over the classes, and the last layer of the network is kept frozen.

- **Continual Test Time Adaptation (CoTTA)** [48] uses a teacher student approach in combination with augmentations. *Anti-Collapse Mechanism:* Every iteration, 0.1% of the weights are reset back to their pretrained values.

- **Efficient Test Time Adaptation (EATA)** [30] uses 2 weighing functions to weigh its outputs: the first based on their entropy (lower entropy outputs get a higher weight), the second based on diversity (outputs that are similar to seen before outputs are excluded). *Anti-Collapse Mechanism:* An $L_2$ regularizer loss is used to encourage the model's weights to stay close to their initial values.

- **EATA Without Weight Regularization (ETA)** For completeness, we also test ETA, which is EATA but without the regularizer loss, proposed in [30].

- **RDumb** is our proposed baseline to mitigate collapse via resetting. We reset every $T = 1,000$ steps, as determined by a hyperparameter search on the holdout set (Section 6).

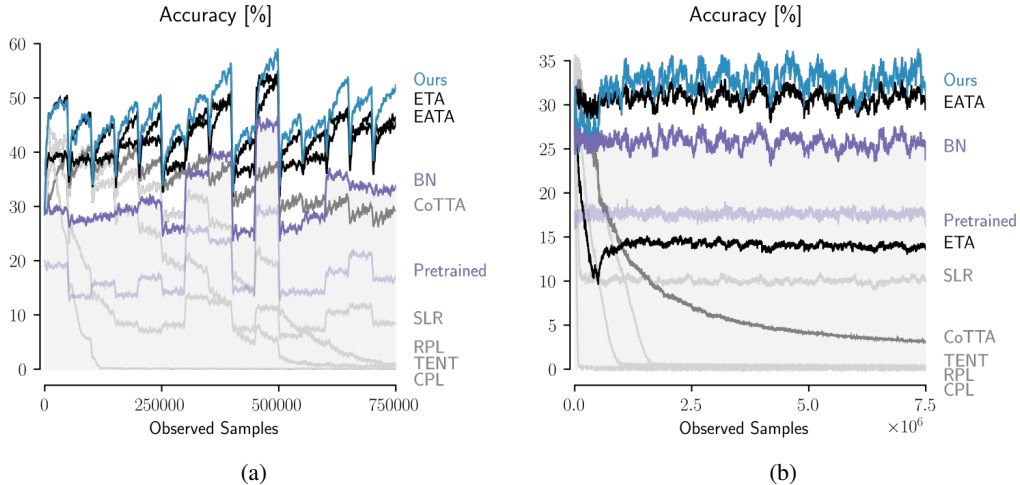

Figure 3: Adaptation performance of all evaluated models depending on the number of observed samples so far. (a) CIN-C. Model performances are averaged over the 10 runs of the benchmark. (b) CCC. Model performances are averaged over the 27 runs of the three difficulty levels. See Appendix C, Figure 10 for separate plots for CCC Easy, Medium and Hard.

Following the original implementations, Tent, ETA, EATA, and RDumb use SGD with a learning rate of $2.5 \cdot 10^{-4}$. RPL uses SGD with a learning rate of $5 \cdot 10^{-4}$. SLR uses the Adam optimizer with a learning rate of $6 \cdot 10^{-4}$. CoTTA uses SGD with a learning rate of 0.01, and CPL uses SGD with a learning rate of 0.001.

## 5 Results

**CCC reveals collapse during continual adaptation, unlike CIN-C.** For three models that were evaluated, evaluation on CIN-C yielded inconclusive or inaccurate results in detecting collapse: CoTTA collapses on CCC, while CIN-C shows it to be on a downward trend, but end performance still outperformed the baseline (Figure 3a). Additionally, ETA shows no signs of collapse on CIN-C, while collapsing very clearly on CCC (Figure 3b, more precisely on CCC-Medium and CCC-Hard, see Appendix Figure 10). When tested using ViT backbones, EATA is better than the pretrained model on CIN-C (Figure 4a), but worse than the pretrained model on CCC (Figure 4b, Table 2,3). Lastly, SLR on CIN-C appears to be somewhat stable, but only at around 10% accuracy. CCC reveals this to be only partly true: on CCC-Hard, SLR is not stable and collapses to nearly chance accuracy.

Table 1: Mean accuracy of ResNet-50 models on CIN-C, CIN-3DCC and CCC. For each CCC split (Easy, Medium, and Hard), a mean of 9 runs is taken. For the CIN-C and CIN-3DCC experiments, the accuracy reported is the mean of 10 different noise permutations. Grey indicates collapse.

| Adaptation method | CIN-C | CIN-3DCC | CCC-Easy | CCC-Medium | CCC-Hard | Average |
| --- | --- | --- | --- | --- | --- | --- |
| Pretrained [10] | $18.0 \pm 0.0$ | $31.5 \pm 0.0$ | $34.1 \pm 0.22$ | $17.3 \pm 0.21$ | $1.5 \pm 0.02$ | 20.5 |
| BN [29, 38] | $31.5 \pm 0.02$ | $35.7 \pm 0.02$ | $42.6 \pm 0.39$ | $27.9 \pm 0.74$ | $6.8 \pm 0.31$ | 28.9 |
| Tent [47] | $15.6 \pm 3.5$ | $24.4 \pm 3.5$ | $3.9 \pm 0.58$ | $1.4 \pm 0.17$ | $0.51 \pm 0.07$ | 9.2 |
| RPL [35] | $21.8 \pm 3.6$ | $30.0 \pm 3.6$ | $7.5 \pm 0.83$ | $2.7 \pm 0.36$ | $0.67 \pm 0.14$ | 12.5 |
| SLR [28] | $12.4 \pm 7.7$ | $12.2 \pm 7.7$ | $22.2 \pm 18.4$ | $7.7 \pm 9.0$ | $0.66 \pm 0.57$ | 11.0 |
| CPL [8] | $3.0 \pm 3.3$ | $5.7 \pm 3.3$ | $0.41 \pm 0.06$ | $0.22 \pm 0.03$ | $0.14 \pm 0.01$ | 1.9 |
| CoTTA [48] | $34.0 \pm 0.68$ | $37.6 \pm 0.68$ | $14.9 \pm 0.88$ | $7.7 \pm 0.43$ | $1.1 \pm 0.16$ | 19.1 |
| EATA [30] | $41.8 \pm 0.98$ | $43.6 \pm 0.98$ | $48.2 \pm 0.6$ | $35.4 \pm 1.0$ | $8.7 \pm 0.8$ | 35.5 |
| ETA [30] | $43.8 \pm 0.33$ | $42.7 \pm 0.33$ | $41.4 \pm 0.95$ | $1.1 \pm 0.43$ | $0.23 \pm 0.05$ | 25.8 |
| RDumb (ours) | $\mathbf{46.5 \pm 0.15}$ | $\mathbf{45.2 \pm 0.15}$ | $\mathbf{49.3 \pm 0.88}$ | $\mathbf{38.9 \pm 1.4}$ | $\mathbf{9.6 \pm 1.6}$ | $\mathbf{37.9}$ |

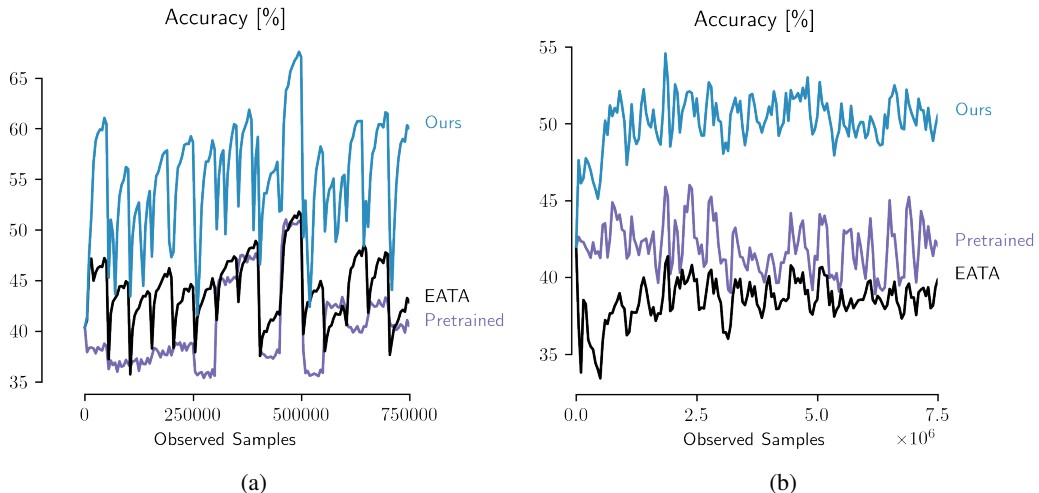

Figure 4: TTA using a ViT backbone: (a) On CIN-C, EATA is better than the pretrained baseline (44.4% points vs 40.1% points). (b) On CCC-Medium, EATA is worse than the pretrained baseline (38.5% points vs 42.0% points). RDumb (ours) is consistently better than both EATA and the baseline.

Table 2: Mean accuracy of different backbone architectures on CCC-Medium. Accuracy reported is an average across 9 runs. Backbones used: [3, 10, 23, 43, 51], †: AugMix [14], ‡: DeepAugment [11]. Grey indicates collapse.

| Method | RN18 | RN34 | RN50 | RN50† | RN50†‡ | RNXt101†‡ | ViT-B16 | MaxViT-T | SwinViT-T |
|---|---|---|---|---|---|---|---|---|---|
| Pretrained | 12.2 | 17.3 | 17.3 | 27.9 | 38.9 | 47.8 | 42.0 | 45.1 | 33.2 |
| EATA | 26.8 | 30.8 | 35.4 | 46.5 | **52.3** | **58.5** | 38.5 | 47.1 | 35.6 |
| RDumb | **32.5** | **37.2** | **38.9** | **47.0** | 51.9 | 58.4 | **50.2** | **49.9** | **36.5** |

In summary, models evaluated on CCC show clear limits, which are impossible to see on CIN-C because of the high difficulty variance between runs, and its short length.

**RDumb is a strong baseline for continual adaptation.** RDumb outperforms all previous methods on both established benchmarks (CIN-C, CIN-3DCC) as well as our continual adaptation benchmark, CCC (Table 1 and Figure 3). Concretely, we outperform EATA and increase accuracy by more than 11% on CIN-C (improving from 41.8% points to 46.5% points), and by almost 7% when averaged all evaluation datasets. While not able to outperform RDumb, we note that EATA is also a strong method for preventing collapse except for the counterexample in Table 2.

**The results transfer to Imagenet-3D Common Corruptions.** To further demonstrate the effectiveness of our method, we show results on Imagenet-3DCC [16], which features 12 types of corruptions, which take the geometry and distances between objects into account when applied to an image. Similarly to CIN-C, we test our models on 10 different permutations of concatenations of all the noises of IN-3DCC, which we call CIN-3DCC. As in the case of CIN-C and CCC, RDumb outperforms all previous methods (Table 1).

**The results transfer to Vision Transformers.** To further validate our claims, we test both EATA and our method with a Vision Transformer (ViT, 3) backbone (Figure 4). The difference in average accuracy between our method and EATA is larger when using a ViT, as compared to a ResNet-50: on CIN-C and CCC the gap is 10.9% points and 11.7% points respectively. Additionally, EATA's accuracy on CCC is below that of a pretrained, non-adapting model[2]. This collapse can only be seen by using CCC, and not when evaluating on CIN-C.

---

[2]Increasing the regularizer parameter value does not help stabilize the model, see Appendix D.

Table 3: Mean accuracy of different backbone architectures on CCC-Hard. Accuracy reported is an average across 9 runs. Backbones used: [3, 10, 23, 43, 51], †: AugMix [14], ‡: DeepAugment [11]. Grey indicates collapse.

| Method | RN18 | RN34 | RN50 | RN50† | RN50†‡ | RNXt101†‡ | ViT-B16 | MaxViT-T | SwinViT-T |
|---|---|---|---|---|---|---|---|---|---|
| Pretrained | 0.82 | 1.3 | 1.5 | 5.6 | 24.3 | 15.6 | **22.0** | **22.0** | **9.3** |
| EATA | 6.4 | 7.6 | 8.7 | **17.7** | **30.3** | **36.3** | 8.6 | 15.4 | 8.6 |
| RDumb | **8.3** | **10.7** | **9.6** | 14.7 | 29.9 | 35.6 | **23.8** | **22.0** | 8.0 |

**RDumb allows adaptation of a variety of architectures without tuning.** We evaluate RDumb and EATA across a range of popular backbone architectures. Out of the nine architectures evaluated (see Table 2,3), RDumb outperformed EATA by an average margin of 4.5% points on seven of them, and worse by an average margin of only 0.25% points on the remaining two.

# 6 Analysis and Ablations

**Optimal reset intervals.** To determine the optimal reset interval, we run ETA with reset intervals $T \in [125, 250, 500, 1000, 1500, 2000]$ on CIN-C using the IN-C holdout noises. We concatenate the 4 holdout noises at severity 5 as our base test set. This base test set is repeated until the model sees 750k images, which is equal to the length of CIN-C. We do this for every permutation of the 4 holdout corruptions. On this holdout set, we find that the optimal $T$ is equal to 1,000.

Table 4: Accuracy of our method for different resetting times on CIN-C-Holdout

| T (steps) | 125 | 250 | 500 | 1000 | 1500 | 2000 |
|---|---|---|---|---|---|---|
| Acc. [%] | 42.1 | 44.4 | 46.0 | **46.7** | 46.5 | 46.4 |

**RDumb is less sensitive to hyperparameters.** An added benefit to our method is that it is less sensitive to hyperparameters than EATA. We conduct a simple hyperparameter search of the $E_0$ parameter—the hyperparameter that controls how many outputs get filtered out because of their high entropy. Our method consistently outperforms EATA across every hyperparameter tested (Table 5), and for the highest value, 0.7, EATA collapses to almost chance accuracy on all splits, while our method does not. In addition, RDumb's performance benefits from finetuning ($H_0 = \{0.2, 0.3\}$), while EATA is not able to improve.

Table 5: Average accuracy on all of CCC splits on a variety of $H_0$ values. For all other experiments in this paper we use $H_0 = 0.4 \times \ln 10^3$, as in [30].

| $H_0 \times \ln 10^3$ | 0.1 | 0.2 | 0.3 | 0.4 | 0.5 | 0.6 | 0.7 |
|---|---|---|---|---|---|---|---|
| EATA | 27.8 | 27.9 | 29.9 | **30.8** | 28.7 | 28.0 | 0.33 |
| RDumb (ours) | 31.6 | 32.9 | **33.1** | 32.6 | 30.7 | 25.7 | 16.8 |

**RDumb is effective because ETA reaches maximum adaptation performance fast.** ETA is quick to adapt to new noises from scratch. On each of the holdout set noises and severities, ETA reaches its maximum accuracy after seeing only about 12,500 samples, which is about 200 adaptation steps (Figure 5a). After that, accuracy decays at a pace slower than its initial increase. Therefore, when resetting and readapting from scratch, only a few steps with substantially suboptimal predictions are encountered before performance is again close to optimal.

**Comparing resetting to regularization.** Previous works typically optimize two loss terms: one loss encourages adaptation on unseen data, another loss regularizes the model to prevent collapse. Having to optimize two losses should be harder than optimizing just one – we see evidence for this

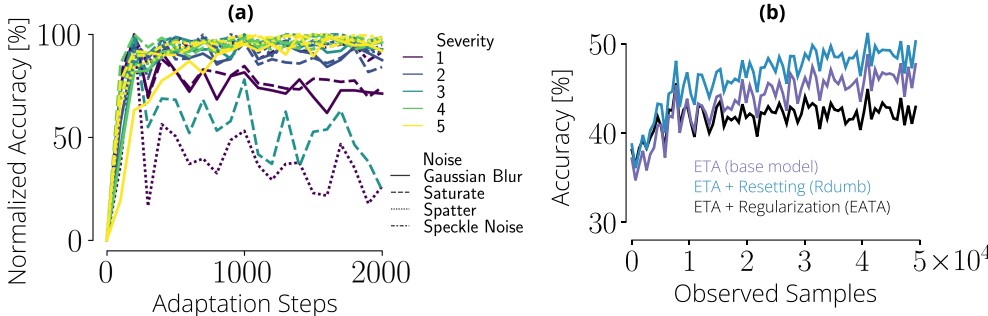

Figure 5: (a) ETA's normalized accuracy over time, on the ImageNet-C holdout noises and each of their severities. For every noise in the holdout set, ETA reaches its maximum accuracy very quickly. (b) Rdumb shares ETA's property of fast adaptation, while regularization in EATA slows adaptation.

in short term adaptation on CIN-C (Figure 5b): ETA and EATA optimize the same loss, but EATA additionally optimizes an anti-collapse loss. Consequently, ETA beats EATA by 2% points on CIN-C.

**Collapse Analysis.** We now investigate potential causes and effects of the observed collapse behavior. We propose a theoretical model, fully specified in Appendix A, which can explain both collapsing and non-collapsing runs. The model consists of a batch norm layer followed by a linear layer trained with the Tent objective. Within this model, we can present two scenarios. In the first, the model successfully adapts and plateaus at high accuracy (Figure 7a). In the second, we see early adaptation which is then followed by collapse (Figure 6a,7a). The properties of noise in the data influence whether we observe the case of successful or unsuccessful adaptation.

Interestingly, the model predicts that the magnitude of weights increases over the course of optimization; this signature of entropy minimization can be found in both the theoretical model and a real experiment using RDumb without resetting on CCC-Medium (Figure 6). Unfortunately, weight explosion happens only *after* model performance is already collapsed (Figure 8b). The effect is observable across all layers (Figure 6c,8).

## 7 Discussion and Related Work

**Domain Adaptation.** In practice, the data distribution at deployment is different from training, and hence the task of *domain adaptation*, i.e., the task of adapting models to different target distributions has received a lot of attention [6, 14, 21, 22, 36, 42, 47]. The methods on domain adaptation split into different categories based on what information is assumed to be available during adaptation. While some methods assume access to labeled data for the target distribution [27, 53], *unsupervised domain adaptation* methods assume that the model has access to labeled source data and unlabeled target data at adaptation time [5, 21, 22, 41]. Most useful for practical applications is the case of *test-time adaptation*, where the task is to adapt to the target data on the fly, without having access to the full target distribution, or the original training distribution [29, 30, 35, 38, 42, 47, 54].

In addition to the division made above, one can further distinguish what assumptions are made about how the target domain is changing. Many academic benchmarks focus on one-time distribution shifts. However, in practical applications, the target distribution can easily change perpetually over time, e.g., due to changing weather and lightness conditions, or due to sensor corruptions. Therefore, the latter setting of *continual adaptation* has been receiving increasing attention recently. The earliest example of adapting a classifier to an evolving target domain that we are aware of is [15], which learn a series of transformations to keep the data representation similar over time. [5, 44, 49] use an adversarial domain adaptation approach for this. [2] pointed out that two of these approaches can be prone to catastrophic forgetting [34]. To deal with this, different solutions have been proposed [1, 2, 24, 28, 30, 48].

Test-time adaptation methods have classically been applied in the setting of one-time domain change, but can be readily applied in the setting of continual adaptation, and some recent methods have been explicitly designed and tested with continual adaptation in mind [30, 47, 48]. Because TTA methods use only test data and don't alter the training procedure, they are particularly easy to apply and have

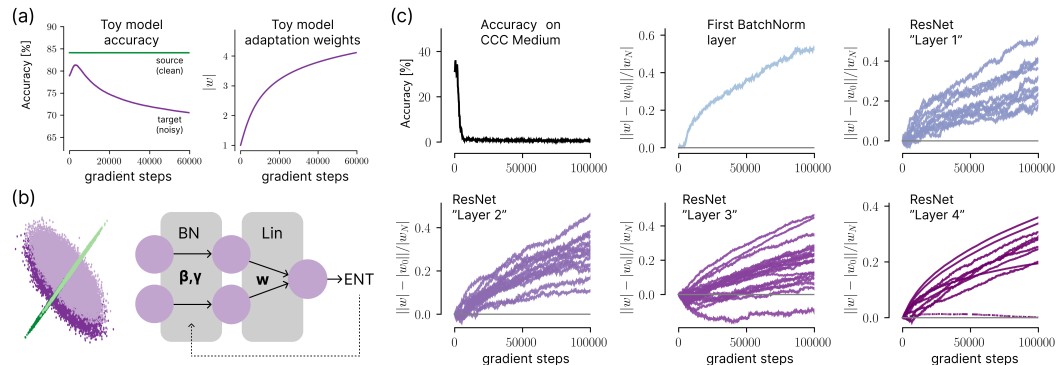

Figure 6: Analysis of entropy minimization collapse on synthetic and real data. Learning dynamics in terms of accuracy and weight magnitude are shown in (a) for a two layer toy model consisting of batch norm and linear layer (b). Consistent with the theoretical analysis, we find that the adaptation weights in all layers increase over time continually (c), even long after the collapse as indicated by Accuracy on CCC-Medium has happened. Refer to Figure 7–8 and Appendix A for additional properties of the toy model (a–b) and a zoomed-in view on (c).

been shown to be superior to other domain adaptation approaches [6, 29, 36, 38], Therefore, we focus only on TTA methods, which we discussed in more detail in Section 4.

**Continual Adaptation Benchmarks.** While continually changing datasets are used in the continual learning literature, e.g. [4, 9, 25, 39, 40, 52], they have been used in TTA benchmarks only very recently. In contrast to all previous benchmarks, we want to evaluate how continual adaptation methods change over long periods of time, when the noise changes in a continuous manner. The longest datasets for TTA were made up of hundreds of thousands of labeled images in total, while we adapt to 7.5M images per run. Other datasets are comprised of short video clips [25, 39, 40] 10-20 seconds in length. Besides maximizing its length, we set out to create a dataset that is well calibrated and closely related to the well-known ImageNet-C dataset. Additionally, with our noise synthesis, we can guarantee a wide variety of noises in each one of our evaluation runs, we can control the speed at which the noise changes, and we can control the difficulty of the generated noise. Lastly, CCC accounts for different adaptation speeds, as demonstrated by [35] and [28]. They showed that training their methods on ImageNet-C for more than one epoch leads to better performance.

## 8 Conclusion

TTA techniques are increasingly applied to continual adaptation settings. Yet, we show that all current TTA techniques collapse in some continual adaptation settings, eventually performing worse than even non-adapting models. And while some methods are stable in some situations, they are still outperformed by our simplistic baseline "RDumb", which avoids collapse by resetting the model to its pretrained state periodically. These observations were made possible by our new benchmark for continual adaptation (CCC), which was carefully designed for the precise assessment of long and short term adaptation behaviour of TTA methods and we envision it to be a helpful tool for the development of new, more stable adaptation methods.

## Acknowledgements

We thank Evgenia Rusak, Çağatay Yıldız, Shyamgopal Karthik, and Ofir Press for helpful discussions and feedback on the manuscript.

We thank the International Max Planck Research School for Intelligent Systems (IMPRS-IS) for supporting OP and StS; StS acknowledges his membership in the European Laboratory for Learning and Intelligent Systems (ELLIS) PhD program. StS was supported by a Google Research PhD Fellowship. MB is a member of the Machine Learning Cluster of Excellence, EXC number 2064/1 – Project No 390727645 and acknowledges support by the German Research Foundation (DFG): SFB 1233, Robust Vision: Inference Principles and Neural Mechanisms, TP 4, Project No: 276693517. This work was supported by the Tübingen AI Center. The authors declare no conflicts of interests.

## Author contributions

Author order below is determined by contribution among the respective category. Conceptualization, RDumb: OP with input from all authors; Conceptualization, CCC: StS, MB, MK; Methodology: OP, StS; Software: OP; Data Curation: OP; Investigation: OP; Formal analysis: MK, StS, OP; Visualization: StS, OP with input from all authors; Writing, Original Draft: OP, StS, MK; Writing, Review & Editing: all authors; Supervision: MB, MK, StS.

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

# A 2D Example Experiments and Analysis

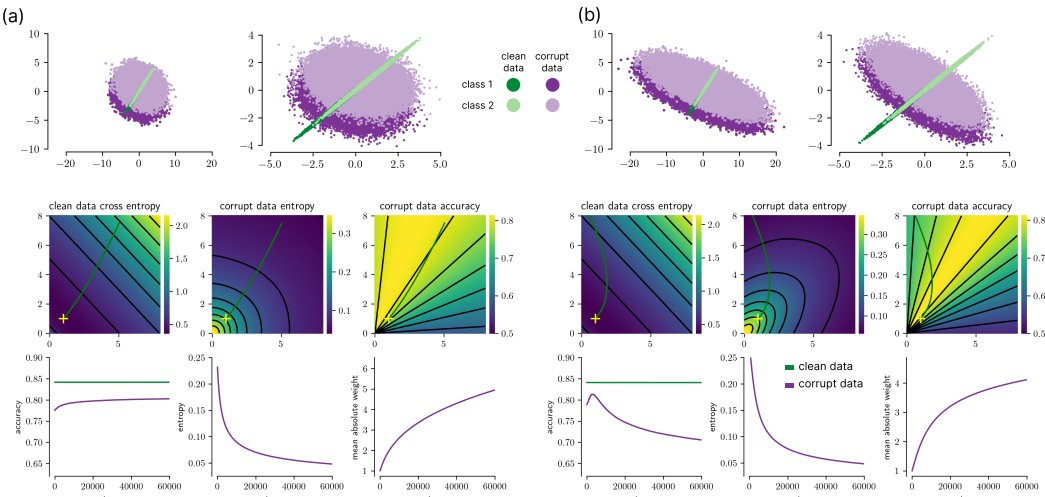

Figure 7: Theoretical analysis of adaptation under distribution shift. Collapsing or non-collapsing behavior of entropy minimization can be reproduced with a simple 2d Gaussian binary classification example, a domain shift which slightly rotates the data and adds Gaussian noise, and a model which consists of a batch norm layer followed by logistic regression. **Top:** clean and corrupt data for two classes before (a) and after (b) batch norm. **Middle:** learning dynamics of entropy minimization in the 2d adaptation parameter space starting from initial parameters (yellow marker) over time. **Bottom:** accuracy, entropy, and size of adaptation weights over time.

In order to better understand collapse, we constructed a simple 2D Gaussian binary classification example.

**Data.** The 2D datasets are constructed as follows: the data is sampled by drawing sampled from two Gaussian blobs with identical variance corresponding to the two classes $\mathcal{N}(\mu_i, \Sigma)$. For the corrupted case, the data is then rotated by an angle $\theta$ and combined with additional additive Gaussian noise $\mathcal{N}(0, \Sigma_{\text{corrupt}})$. Finally, both in the clean and the corrupt data case, the data is rotated by an angle of $-\frac{\pi}{4}$ (which minimizes the effect of the batch normalization in the model).

$$Y \sim \text{Ber}(0.5) \tag{2}$$

$$X \sim \mathcal{N}(\mu_Y, \Sigma) \tag{3}$$

$$X_{\text{clean}} = R_{-\frac{\pi}{4}} X \tag{4}$$

$$X_{\text{corrupt}} \sim \mathcal{N}(R_{\theta_{c_1}} X_{\text{clean}}, R_{-\frac{\pi}{4}}^{\top} R_{\theta_{c_2}}^{\top} \tilde{\Sigma} R_{\theta_{c_2}} R_{-\frac{\pi}{4}}) \tag{5}$$

$$\text{with } \Sigma = \begin{pmatrix} \sigma_1 & 0 \\ 0 & \sigma_2 \end{pmatrix}, \quad \sigma_1 \gg \sigma_2 \tag{6}$$

$$\tilde{\Sigma} = \begin{pmatrix} \tilde{\sigma}_1 & 0 \\ 0 & \tilde{\sigma}_2 \end{pmatrix}, \quad \tilde{\sigma}_2 \geq \tilde{\sigma}_1 \tag{7}$$

$$R_\theta = \begin{pmatrix} \cos(\theta) & \sin(\theta) \\ -\sin(\theta) & \cos(\theta) \end{pmatrix} \tag{8}$$

Note that for the clean data, we allocate most of the variance to the class dimension, $\sigma_1$. On the corrupted data, we add noise primarily perpendicular to the class dimension ($\tilde{\sigma}_2$), which is the main defining factor whether or not we observe collapsing behavior. $\theta_{c_1} \neq \theta_{c_2}$ serves to break the symmetry between signal and noise in the data, which results in Tent starting to deviate towards the noise direction.

In our example, we parametrize this model as follows: In condition 1, $\mu_i = (\pm 1, 0)$ and $\Sigma^2 = \begin{pmatrix} 1 & 0 \\ 0 & 0.03 \end{pmatrix}$, $\theta = -\frac{\pi}{9}$ and $\Sigma_{\text{corrupt}}^2 = R_{\theta_c}^T \begin{pmatrix} 0.25 & 0 \\ 0 & 4 \end{pmatrix} R_{\theta_c}$, where $R_{\theta_c}$ is the rotation matrix for an angle of

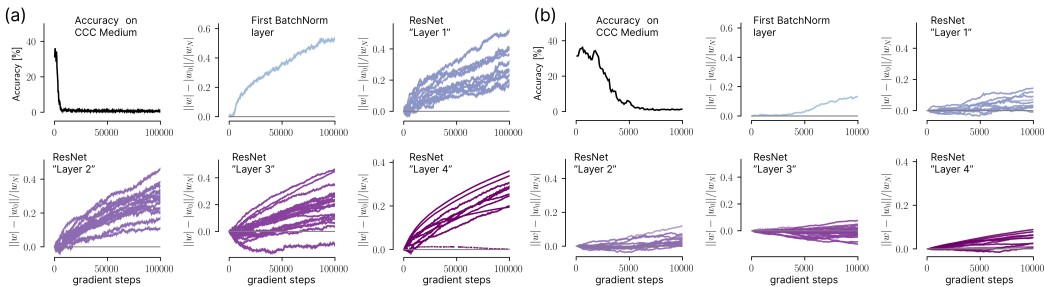

Figure 8: Analysis of entropy minimization collapse on real data. (a) Consistent with the theoretical analysis in Fig. 7, we find that the adaptation weights in all layers increase over time continually, even long after the collapse as indicated by Accuracy on CCC-Medium has happened. (b) shows a zoomed-in view where this increase is not yet apparent, well after the collapse.

$\theta_c = -\frac{\pi}{6}$. In condition 2, everything is the same except for $\Sigma^2_{\text{corrupt}} = R^T_{\theta_c} \left( \begin{smallmatrix} 0.25 & 0 \\ 0 & 25 \end{smallmatrix} \right) R_{\theta_c}$. In both conditions, we sample 300 000 data points which are equally distributed over both classes.

**Model.** The classification model consumes the dataset of shape $N \times 2$ and consists of a batchnorm layer ($\epsilon = 0$) followed by a fully connected layer with two input channels, one output channels, no bias term and a sigmoid nonlinearity. Because our data is always centered, we do not learn the offset parameter of the affine adaptation in the batchnorm layer, but only the scale parameter.

**Model training.** The model is trained to minimize the binary cross-entropy on the clean data. We use batch gradient descent on the whole 300,000 sample dataset with a learning rate of 0.1 and no momentum. We decay the learning rate by a factor of 0.1 after 1000 and 2000 steps and stop training after 3000 steps.

**Model Learning and Adaptation.** We adapt the model on the corrupted data using Tent. More precisely, we optimize the scale parameter of the batch norm layer to minimize the entropy of the predictions using SGD with a learning rate of 0.01 and no momentum. We process the whole dataset in one batch and adapt for 60 000 steps.

**Results.** In the toy model, simple cases emerge where the loss does not result in collapse, or vice versa (Fig. 7a and Fig. 7b, respectively), mainly depending on the relation of signal and noise variances and directions.

The toy example furthermore predicts that the adapted parameters of a model should grow on the long run and indeed we were able to find exactly this effect when running ETA on a ResNet50 on CCC-Medium (Fig. 8), suggesting that our minimal setup successfully reproduces the relevant aspects of the large scale case. However, the weight explosion becomes apparent only after the collapse happens, hence weight regularization is not enough to avoid the collapse (Fig. 8(b)).

In the Fig. 7(a) example, we find the direction of high target domain performance and stay there; in this case entropy minimization is stable. In the Fig. 7(b) experiment, the domain shift adds more noise nearly orthogonal to the signal direction, which entropy minimization tries to use for making high confidence predictions: we still initially find the direction of high target domain performance, but traverse this region and continue into a direction of low entropy and low accuracy. This shows that even in a linear example, entropy minimization can show initial performance improvement and then collapse.

## B  Path Finding Algorithm

Algorithm 1 describes the pseudo code of the algorithm used to generate CCC. The algorithm is based on a set of Calibration Matrices, similar to the one shown in Figure 2. There exists a matrix $m$ for every $(n_a, n_b)$ pair, such that $m[i][j]$ is equal to accuracy of a pretrained ResNet-50 on the

comination of noises $(n_a, n_b)$, and severities $(i/5, j/5)$. We will release the full set of matrices upon publication.

Additionally, Algorithm 2 uses a function *MinValidPath*$(s_1, s_2)$: this function returns the minimum path that starts at $(s_1, s_2)$ and ends at $(0, s_j)$ for some $s_k$. The cost of a path is simply the average of all entries along the path. The minimum path is defined as the path with a cost closest to $b_a$ in absolute terms. Lastly, a path is only valid if it starts with $s_2$ equal to 0, every transition either decreases $s_1$ by 0.25, or increases $s_2$ by 0.25, and stops once $s_1$ is equal to 0.

---

**Algorithm 1** Algorithm used to generate each split of CCC

---

**Require:** $ba, k, T$                   ▷ Baseline accuracy, transition speed, and total split size.
1: $t = 0$                     ▷ Initialize the total images generated counter
2: $c_1, c_2 \sim \mathrm{Uniform}(\{1 \dots 15\})$           ▷ Initialize the first two corruptions.
3: path $\leftarrow \mathrm{CalculatePath}(c_1, c_2, ba)$ ▷ Calculate path along the noise pair with an average accuracy closest to $ba$.
4: **loop**
5:      $s_1, s_2 \leftarrow \mathrm{path}[p]$
6:      Subset $\sim \mathrm{Uniform}(\mathrm{ImageNetVal})$
7:      apply $(c_1, s_1, c_2, s_2)$ to Subset
8:      save Subset
9:      $t \leftarrow k$
10:      **if** $t >= T$ **then return**
11:      **end if**
12:      **if** $p = \mathrm{len}(\mathrm{path}) - 1$ **then**
13:          $c_1 \leftarrow c_2$
14:          $c_2 \sim \mathrm{Uniform}(\{1 \dots 14\})$
15:          path $\leftarrow \mathrm{CalculatePath}(c_1, c_2, ba)$          ▷ Calculate new path along the new noise pair.
16:          $p \leftarrow 0$
17:      **else**
18:          $p \leftarrow p + 1$          ▷ Move to the next severity combination
19:      **end if**
20: **end loop**

---

**Algorithm 2** CalculatePath

---

**Require:** $c_1, c_2, ba$                   ▷ The 2 noises, and the baseline accuracy
1: $m \leftarrow \mathrm{CalibrationMatrix}[c_1][c_2]$
2: $\mathrm{MinPath}, \mathrm{MinCost} \leftarrow \mathrm{MinValidPath}(0, 0, m, ba)$
3: **for** $s_1 \in 0.25, 0.5, 0.75, \dots, 5$ **do**
4:      $m \leftarrow \mathrm{MinValidPath}(1, 0, m, ba)$
5:      $\mathrm{MinPath}, \mathrm{MinCost} \leftarrow \mathrm{MinValidPath}(s_1, 0, m, ba)$
6: **end for**
7: **return** $MinPath$

---

The resulting dataset features transitions between two different noises like the ones seen in Figure 9 (a). We additionally plot the accuracy of a pretrained ResNet-50, alongisde the severities of the different noises in Figure 9 (b).

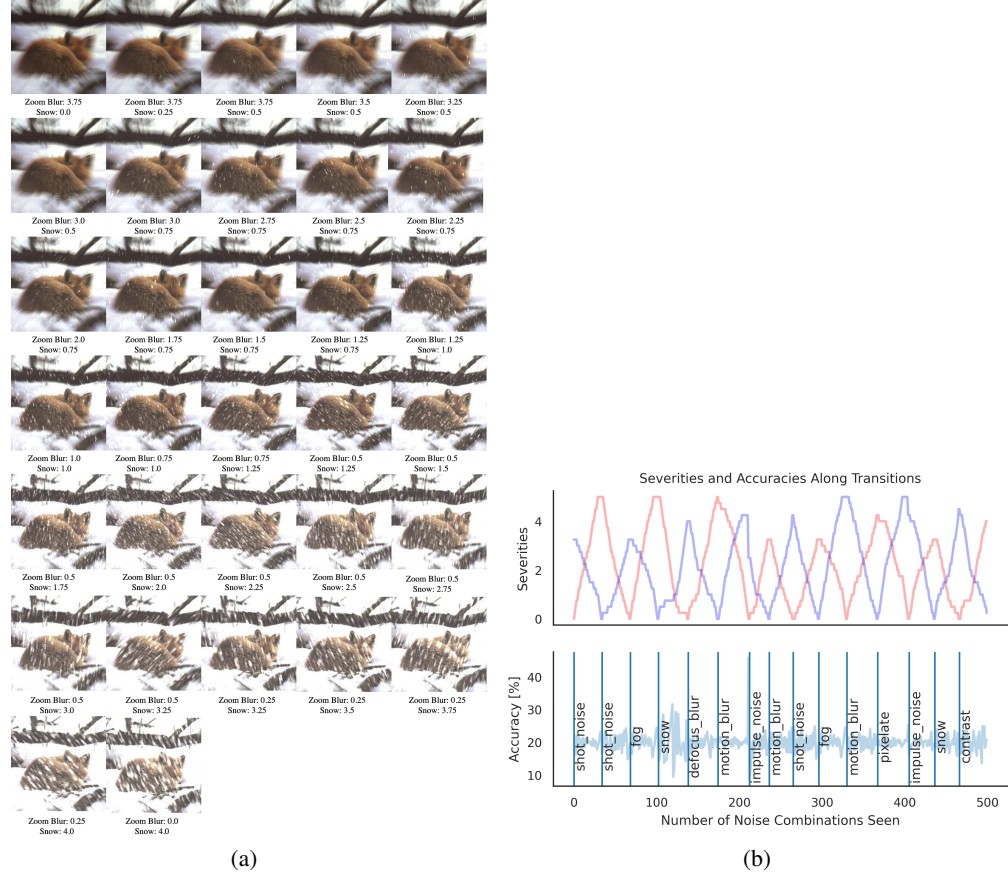

(a)          (b)

Figure 9: **(a)** Visualization of CCC-Medium's smooth transition between Zoom Blur to Snow. Note: CCC additionally uses random flips and crops, which are not shown here. **(b)** As CCC transitions between noises, the severities of the first noise (red), and the second noise (blue) go up and down correspondingly, in order to keep the accuracy of a pretrained model stable.

We additionally share the following metadata about the length of traversals in CCC-Easy/Medium/Hard:

Table 6: CCC traversal length statistics, for each CCC split.

|  | Min | Max | Mean | Median |
|---|---|---|---|---|
| CCC-Easy | 11 | 36 | 22.8 | 23 |
| CCC-Medium | 21 | 41 | 33.9 | 34 |
| CCC-Hard | 41 | 41 | 41 | 41 |

## C    CCC Plots

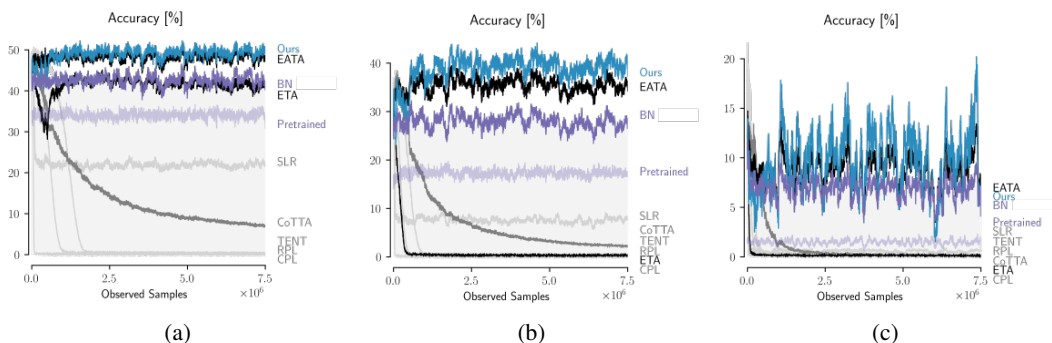

Figure 10: Adaptation performance of all evaluated models using a ResNet-50 backbone. (a) CCC Easy. (b) CCC Medium. (c) CCC Hard. For all subplots, model performances are averaged over the 9 runs of the respective difficulty level.

## D    EATA Implementation and Ablations

Our implementation of EATA differs from the official implementation[3]. The reason for this is that the official implementation uses clean ImageNet validation images to calculate the Fisher vector matrix for its regularizer[4]. This stands in contradiction with the method, which should not have access to the training distribution at test time, as shown in the paper in Table 1.

Instead of using $2,000$ ImageNet validation images, we calculate the Fisher matrix using the first $2,000$ images in our data stream. We conduct a hyperparameter search on the weight regularizer tradeoff parameter $\beta$:

| $\beta$ | 25 | 50 | 100 | 250 | 500 | 1000 | 1500 | 2000 |
|---|---|---|---|---|---|---|---|---|
| Acc. [%] | 46.5 | 46.9 | **47.1** | 46.7 | 46.1 | 45.6 | 44.8 | 44.0 |

Table 7: Accuracy of EATA on CIN-C holdout noises for different values of the weight regularizer loss.

Using the optimal value, 100 led to worse results than the default value, 2000, on CCC:

|  | CIN-C | CCC-Easy | CCC-Medium | CCC-Hard | CCC Avg |
|---|---|---|---|---|---|
| EATA-100 | 46.7 | 47.7 | 36.5 | 3.8 | 29.3 |
| EATA-2000 | 41.8 | 48.2 | 35.4 | 8.7 | 30.8 |
| Ours | 46.5 | 49.3 | 38.9 | 9.6 | 32.6 |

Table 8: Accuracy of EATA on CIN-C holdout noises for different values of the Fisher alpha.

In the end, we used the original value of 2000, as that was optimal on the CCC dataset.

In addition, we conducted a hyperparameter search for EATA on a ViT backbone. As shown in 4, EATA performs worse than a pretrained, non adapting baseline in this setting. To that end, we tried to stabilize the model by increasing the value of $\beta$, the hyperparameter that controls the weight of the anti-forgetting regularizer. As with the previous experiment, the original value of 2000 is optimal.

---

[3] https://github.com/mr-eggplant/EATA, version f739b3668c
[4] https://github.com/mr-eggplant/EATA/blob/f739b3668c/main.py#L144

| $\beta$ | 2000 | 3000 | 4000 |
|---|---|---|---|
| Acc. [%] | **38.5** | 27.8 | 16.5 |

Table 9: Accuracy of EATA on CCC-Medium using a ViT backbone, for different values of the regularizer, $\beta$.

# E   Novelty of Resetting

Our work is the first to propose resetting to solve collapse in TTA methods. Notably, while prior work [30, 47, 54] has briefly touched upon the concept of episodic resetting, the methodology and its application is significantly distinct and unrelated to collapse in TTA.

- **Tent** [47] mentions episodic in the context of overfitting to a single sample in segmentation (similar to [54]'s overfitting to a single sample). Resetting here is unrelated to collapse, as the paper doesn't discuss collapse at all.

- Although **MEMO** [54] uses resetting, it does so because it overfits to one image (and its augmentations) every step. MEMO doesn't discuss collapse or catastrophic forgetting. MEMO compares itself to a version of Tent that resets after every step (which they call Tent + episodic resetting), because MEMO without augmentations is similar to Tent + episodic resetting with a batch size of 1. (Note: MEMO is outperformed by BN/Tent/ETA when using the standard batch size of 64)

- **EATA** [30] shows results for Tent + episodic resetting after every step in its tables, but provides no reasoning or discussion for doing this. Tent + episodic resetting is outperformed by regular Tent.

# F   CIFAR10 Experiments

We conduct CIFAR10 experiments to show the need for ImageNet scale benchmarks. Tent, without an anti-collapse mechanism, does not collapse on CIFAR10-C, even after seeing 100 million images Fig. 11a,b. Like ImageNet, CIFAR10-C's noises also exhibit high variance in difficulty Fig. 11c.

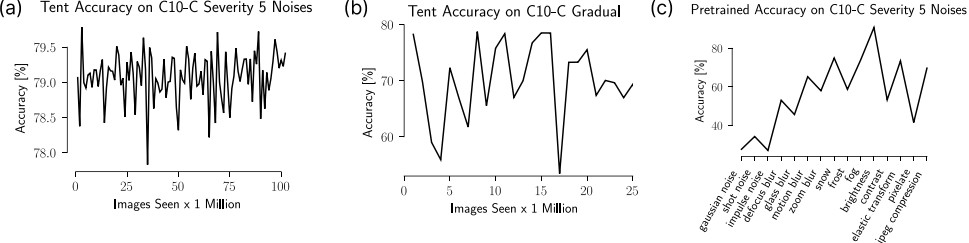

Figure 11: **(a)** When tested on an infinite concatenation of severity 5 noises, Tent does not collapse even after seeing 100M CIFAR scale images. **(b)** Tent does not collapse to chance level when tested on a long term variant of CIFAR10-C gradual. **(c)** CIFAR10-C exhibits great variations between individual corruptions, similar to ImageNet-C.

# G   Compute details

We conduct all experiments on Nvidia RTX 2080 TI GPUs with 12GB memory per device. All experiments except our study on larger models were conducted on a single GPU. For CoTTA experiments, we use data parallel training on 2 GPUs. A bulk of the compute spent for this work was on computing baseline accuracies on the calibration dataset, which contains 463M images.

# H Software and Dataset Licenses

## H.1 Datasets

- ImageNet-C [12]: Creative Commons Attribution 4.0 International,
  https://zenodo.org/record/2235448
- ImageNet-C [12], code for generating corruptions: Apache License 2.0
  https://github.com/hendrycks/robustness
- ImageNet-3D-CC [16]: CC-BY-NC 4.0 License
  https://github.com/EPFL-VILAB/3DCommonCorruptions

## H.2 Models

- PyTorch's [31] Backbones
  https://pytorch.org/vision/stable/models.html
- Adaptive BN [29, 38]:
  Apache License 2.0, https://github.com/bethgelab/robustness
- Tent [47]: MIT License,
  https://github.com/DequanWang/tent
- RPL [35]: Apache License 2.0,
  https://github.com/bethgelab/robustness
- CoTTA [48]: MIT License,
  https://github.com/qinenergy/cotta
- CPL [8]: MIT License,
  https://github.com/locuslab/tta_conjugate
- EATA [30]: MIT License
  https://github.com/mr-eggplant/EATA

