# OpenReview forum: "RDumb: A simple approach that questions our progress in continual test-time adaptation"
_NeurIPS.cc/2023/Conference — NeurIPS 2023 poster_

### Official Review · Reviewer_Gw5H · 2023-07-05

**Soundness:** 4 excellent
**Presentation:** 3 good
**Contribution:** 4 excellent
**Rating:** 7
**Confidence:** 3

**Summary:**

Test-time adaptation techniques seek to adapt a model to new unlabelled samples without access to the original training data. However, TTA approaches are typically not evaluated for long runtimes. This work proposes a new benchmark Continuously Changing Corruptions (CCC), which is a stream of corrupted ImageNet images with gradual changes and varying degrees of difficulty. Previous approaches fail on (CCC) when compared to the pretrained model as well as a simple new baseline method dubbed RDumb. RDumb is shown to achieve high performance across several experimental settings.

**Strengths:**

Strengths:
- The overall message of the paper is easy to follow and the experiments are illustrative of a serious shortcoming in the evaluation of TTA methods.
- The baseline is simple and satisfies all the TTA assumptions, and should be considered as a baseline in future works in this area
- Experiments are very extensive and thorough

**Weaknesses:**

Weaknesses:
- There are a few issues related to clarity that should be addressed in the next version of the paper. See Questions
- I would encourage the authors to add error bars to their tables, so that it is clear that the differences are statistically significant



**Questions:**

Questions:
1) The mathematical notation in Eq. 1 is a bit difficult to parse. In particular, it is unclear what is meant by $cos(y_t, \bar{y}_t)$. Is this cosine similarity or cosine distance? I would encourage the authors to clarify this and also provide a clear description of the equation in the text.

2) Can the authors precisely define "collapse"? This term is used but is never defined.

3) In Figures 3 and 4, if RDumb is being reset to the pretrained model weights, then why do the blue and purple lines never intersect?

4) It would be interesting to test CLIP image encoders [1] that are naturally robust and have high zero-shot accuracy on OOD samples. In future work, I would be curious to see how such models behave on the proposed CCC when TTA techniques are applied.

References:
- Learning Transferable Visual Models From Natural Language Supervision, Radford et al. 2021

**Limitations:**

Limitations are addressed.

---

> ### Author Rebuttal · Authors · 2023-08-10
>
> Thank you for your review.  We’re happy you found our paper to be easy to follow, and the experiments to be extensive. We want to address your questions:
> > I would encourage the authors to add error bars to their tables, so that it is clear that the differences are statistically significant
>
> Good point. We updated the paper with a new version of the table:
>
> | Adaptation method    	| CIN-C      	| CIN-3DCC   	| CCC-Easy   	| CCC-Medium 	| CCC-Hard   	| Average     	|
> |--------------------------|----------------|----------------|----------------|----------------|----------------|-----------------|
> | Pretrained (He et al.) | 18.0 $\pm$ 0.0 | 31.5 $\pm$ 0.0 | 34.1 $\pm$ 0.22 | 17.3 $\pm$ 0.21 | 1.5 $\pm$ 0.02 | 20.5        	|
> | BN (Schneider et al.) | 31.5 $\pm$ 0.02 | 35.7 $\pm$ 0.02 | 42.6 $\pm$ 0.39 | 27.9 $\pm$ 0.74 | 6.8 $\pm$ 0.31 | 28.9        	|
> | Tent (Wang et al.) | 15.6 $\pm$ 3.5 | 24.4 $\pm$ 3.5 | 3.9 $\pm$ 0.58 | 1.4 $\pm$ 0.17 | 0.51 $\pm$ 0.07 | 9.2   |
> | RPL (Rusak et al.) | 21.8 $\pm$ 3.6 | 30.0 $\pm$ 3.6 | 7.5 $\pm$ 0.83 | 2.7 $\pm$ 0.36 | 0.67 $\pm$ 0.14 | 12.5 |
> | SLR (Mummadi et al.) | 12.4 $\pm$ 7.7 | 12.2 $\pm$ 7.7 | 22.2 $\pm$ 18.4 | 7.7 $\pm$ 9.0 | 0.66 $\pm$ 0.57 | 11.0 |
> | CPL (Goyal et al.) | 3.0 $\pm$ 3.3 | 5.7 $\pm$ 3.3 | 0.41 $\pm$ 0.06 | 0.22 $\pm$ 0.03 | 0.14 $\pm$ 0.01 | 1.9 |
> | CoTTA (Wang et al.) | 34.0 $\pm$ 0.68 | 37.6 $\pm$ 0.68 | 14.9 $\pm$ 0.88 | 7.7 $\pm$ 0.43 | 1.1 $\pm$ 0.16 | 19.1 |
> | EATA (Niu et al.) | 41.8 $\pm$ 0.98 | 43.6 $\pm$ 0.98 | 48.2 $\pm$ 0.6 | 35.4 $\pm$ 1.0 | 8.7 $\pm$ 0.8 | 35.5 |
> | ETA (Niu et al.) | 43.8 $\pm$ 0.33 | 42.7 $\pm$ 0.33 | 41.4 $\pm$ 0.95 | 1.1 $\pm$ 0.43 | 0.23 $\pm$ 0.05 | 25.8 |
> | RDumb (ours) | **46.5 $\pm$ 0.15** | **45.2 $\pm$ 0.15** | **49.3 $\pm$ 0.88** | **38.9 $\pm$ 1.4** | **9.6 $\pm$ 1.6** | **37.9** |
>
>
>
> > The mathematical notation in Eq. 1 is a bit difficult to parse. In particular, it is unclear what is meant by $\cos(y_t, \hat{y}_t)$. Is this cosine similarity or cosine distance? I would encourage the authors to clarify this and also provide a clear description of the equation in the text.
>
> We apologize for not making this clear - what was meant is cosine similarity. We made this clearer in the text.
>
> >Can the authors precisely define "collapse"? This term is used but is never defined.
>
> We consider a model as collapsed when a model classifies images worse than a pretrained, non-adapting model. Thanks for raising this clarity issue, we will make the term more clear in the next version.
>
>
> >In Figures 3 and 4, if RDumb is being reset to the pretrained model weights, then why do the blue and purple lines never intersect?
>
> Each point on the graph represents an average of 781 iterations (around 50k images seen). This is done so as to not have to plot millions of points. Although RDumb resets, it very quickly becomes better than a pretrained model (see also Figure 5a), which is why the lines don’t intersect.
>
> >It would be interesting to test CLIP image encoders [1] that are naturally robust and have high zero-shot accuracy on OOD samples. In future work, I would be curious to see how such models behave on the proposed CCC when TTA techniques are applied.
>
> We think so too, which is why we think CCC is an important benchmark for future work.
>
> We hope that our response clears up any questions you had, and if it doesn't, we’d be glad to add further clarifications.

---

> > ### Comment · Reviewer_Gw5H · 2023-08-17
> > **Response to Rebuttal**
> >
> > Thank you for the clear rebuttal, all of my concerns are addressed so I have raised my score. I would recommend the authors to address why the lines do not intersect in the next version of the paper.

---

> > > ### Author Response · Authors · 2023-08-21
> > > **Thank you for your response**
> > >
> > > Dear Reviewer,
> > >
> > > Thank you for your responses.

---

### Official Review · Reviewer_xRL5 · 2023-07-06

**Soundness:** 3 good
**Presentation:** 3 good
**Contribution:** 2 fair
**Rating:** 5
**Confidence:** 4

**Summary:**

This paper proposes a new benchmark for continual test-time adaptation (CTTA): CCC (Continually Changing Corruptions).
Experiments for existing state-of-the-art approaches using their proposed benchmark demonstrate that even a non-adapting model performs better than existing approaches for this benchmark.
Further, they propose a simple approach as a baseline, named $\textit{RDumb}$, which periodically resets the model to its pre-trained state, which performs better than the SOTA approaches.

**Strengths:**

1. This paper proposes a more challenging benchmark for CTTA.
2. Combining a pair of corruptions is a novel idea.
3. Experimental results demonstrate that several state-of-the-art approaches perform poorly on this benchmark.
4. A simple periodic reset mechanism with weighted entropy (as in ETA) adaptation loss outperforms the SOTA CTTA approaches.

**Weaknesses:**

1. In CoTTA (Wang et al., CVPR 2022), there is a setting of gradually changing the corruptions (CIFAR10-to-CIFAR10C gradual). So the idea of gradually changing corruption is not entirely new, even though CCC has a combination of two corruptions.
2. This paper does not analyze the reasons for the performance degradation of the SOTA TTA approaches, such as overfitting, error accumulation, catastrophic forgetting, or interference, and does not provide any insights or suggestions on mitigating or preventing it.
3. Experimental results on CIFAR10-to-CIFAR10C, CIFAR100-to-CIFAR100C classification tasks, along with their CCC variants, are missing.

Minor Comment: Line 62: use `` for starting quotes before RDumb

**Questions:**

1. In CoTTA (Wang et al., CVPR 2022), there is a setting of gradually changing the corruptions (CIFAR10-to-CIFAR10C gradual). CoTTA, as well as PETAL [1], reports performance improvement in this gradual setting, so it would be interesting to see how they would work on the CCC variant of CIFAR10-to-CIFAR10C. Do the authors have any intuition behind this?
2. As the authors point out, the ImagenetC dataset is large and other benchmarks are much smaller; the dataset being smaller can also make the task more challenging, since there would be lesser data to learn from for a domain shift. It would be interesting to see how the performance is for CCC variants of CIFAR10-to-CIFAR10C, CIFAR100-to-CIFAR100C.
3. Also, with regards to the semantic segmentation task, do the authors have some idea as to how this dataset/benchmark building mechanism would be applicable?


*References*
1. Brahma, Dhanajit, and Piyush Rai. "A Probabilistic Framework for Lifelong Test-Time Adaptation." Proceedings of the IEEE/CVF Conference on Computer Vision and Pattern Recognition. 2023.

**Limitations:**

1. Since the state-of-the-art approaches are being evaluated on a new benchmark dataset, it is very important to tune their hyperparameters optimally.
2. The details regarding hyperparameter tuning for various SOTA approaches are missing.

---

> ### Author Rebuttal · Authors · 2023-08-10
>
> Thank you for your review. We’re glad you found the dataset novel and challenging, and our experimental results meaningful! We addressed all mentioned weaknesses below and replied to your questions.
>
> > In CoTTA, there is a setting of gradually changing the corruptions [...]. So the idea of gradually changing corruption is not entirely new, even though CCC has a combination of two corruptions.
>
> We would like to point out three important differences and contributions CCC makes over this setting:
>
> First, the setting used in the COTTA paper is way easier than CCC and fails to show a collapse to chance level. We ran an experiment on the concatenated CIFAR10-C dataset for more than 100 million images with no sign of collapse (Fig. 4, Rebuttal PDF). We also run a similar experiments for 25 million images on the gradual CIFAR10 task (Fig. 5). While the fluctuations in performance increase, the setup still fails to produce the collapsing behavior to chance level we observe at ImageNet-scale with CCC. This highlights the difference in difficulty to our “real world sized” CCC dataset.
>
> Second, “gradual” here means a transition from clean to corrupted images. CCC corruptions have a much larger combinatorial space due to the simultaneous application of multiple corruptions.
>
> Third, the C10-C tasks are less-well controlled, which is similar to the issues we discuss for concatenated ImageNet-C. You can see this in Figure 6 in the rebuttal PDF, where performance fluctuates greatly between noises. This makes the dataset not suitable for controlled evaluation of models in a scientific setting.
>
> We are happy to discuss this further and agree with you that we should point out these features of CCC more prominently.
>
> > This paper does not analyze the reasons for the performance degradation of the SOTA TTA approaches, such as overfitting, error accumulation, catastrophic forgetting, or interference, and does not provide any insights or suggestions on mitigating or preventing it.
>
> This is a fair point, and we added an analysis now.
>
> We constructed a simple 2d Gaussian binary classification example, a domain shift which slightly rotates the data and adds Gaussian noise, and a model which consists of a batch norm layer followed by a logistic regression. When running entropy minimization on the adaptation layer of the batch norm, simple cases emerge where the loss does or does not result in collapse, mainly depending on the relation of signal and noise variances and directions (Fig. 1-2).
>
> The toy example furthermore predicts that the adapted parameters of a model should grow on the long run and indeed we were able to find exactly this effect when running ETA on a ResNet50 on CCC-Medium (Fig. 3, rebuttal PDF), suggesting that our minimal setup successfully reproduces the relevant aspects of the large scale case. However, the weight explosion becomes apparent only after the collapse happens, hence weight regularization is not enough to avoid the collapse (Fig 3b).
>
> Thank you for raising this issue, we think that this analysis adds to the paper and will include it in the camera ready version. We would be happy to hear your thoughts on this.
>
> > Experimental results on CIFAR10-to-CIFAR10C, CIFAR100-to-CIFAR100C classification tasks, along with their CCC variants, are missing.
>
> Continual test time adaptations make claims about the performance of real-world scale vision models during deployment time. CIFAR bears none of the desiderata for a dataset to be predictive of model performance in this setup, as we discussed above.
>
> Hence, we refrained from running CIFAR scale experiments, the minimum scale for meaningful evaluation should be ImageNet-C, variants of it, or CCC. These experiments are sufficiently quick to run on typical research hardware.
>
> However, if we missed a good argument on how CIFAR experiments are useful in providing a signal for building continual adaptation models, we are happy to discuss further. In particular, if CIFAR scale experiments would meaningfully change your assessment of our work, we could discuss a suitable setup for hyperparameter search on all models for evaluation (as CoTTA, EATA, etc. lack details on the CIFAR evaluation protocol and how hyperparameter search was performed).
>
> > Also, with regards to the semantic segmentation task, do the authors have some idea as to how this dataset/benchmark building mechanism would be applicable?
>
> While possible with our construction mechanism, building a “Segmentation-CCC” is well beyond of scope for the current work. However, we think that any model should pass the CCC test first before scaled up to even more involved computer vision problems.
>
> Otherwise, we agree that this could be an interesting future work, the limiting factor is the huge compute budget needed to calculate the calibration set.
>
> > Since the state-of-the-art approaches are being evaluated on a new benchmark dataset, it is very important to tune their hyperparameters optimally.
>
> We agree, and we performed hyperparameter searches on both EATA (Appendix C) and CoTTA (see reply to reviewer h5ar).For CoTTA, we ran a hyperparameter search on CoTTA using the same protocol we used for RDumb (Section 6), and found that CoTTA still collapses on every level of CCC.
> Please note that the only additional hyperparam we consider is the resetting interval. All other parameters in RDumb are taken from previous work.
> If you want us to run any other hyperparam search, we would be more than happy to do so, but we doubt it will change our key results (as we saw in CoTTA and EATA).
>
> > The details regarding hyperparameter tuning for various SOTA approaches are missing.
>
> Please see our reply above, and also Section 6 and App. C in the paper. We are happy to add additional information.
> We fully agree with your sentiment of the importance of hyperparameter tuning in TTA, but feel that the practice in our paper matches or exceeds the practiced standards in the field.
> We are happy to discuss further, though.

---

> > ### Comment · Reviewer_xRL5 · 2023-08-15
> > **Some of the concerns addressed with additional experiments**
> >
> > Dear authors,
> >
> > I want to thank the authors for responding to my comments and questions.
> > Some of the questions asked have been addressed.
> >
> > The experiment with toy 2d Gaussian binary classification is interesting.
> >
> > However, the smaller benchmark dataset, as in CIFAR10-to-CIFAR10C, and CIFAR100-to-CIFAR100C, can also make the task more challenging since there would be less data to learn from for a domain shift. This point has not been addressed. Experimental results on these benchmarks would make the case stronger for the proposed approach.
> >
> > Based on the response, I do not have any other queries at this point in time.
> >
> > Thank you.

---

> > > ### Author Response · Authors · 2023-08-17
> > > **Addressing your remaining questions**
> > >
> > > Dear reviewer xRL5,
> > >
> > > Thanks a lot for getting back to us. We are happy that you also find the 2D example interesting, and will add it to the paper. Thanks again for pushing in the direction to add analysis, we think this improves the paper a lot.
> > >
> > > We would like to address the remaining questions now. In summary, we now show that RDumb significantly outperforms CoTTA on both CIFAR10-C and CIFAR100-C, plus add some further discussion on CIFAR below. Thanks for suggesting these experiments which now further corroborate our claims.
> > >
> > > In more detail, we would first like to clarify a possible misunderstanding, as you write:
> > >
> > > > since there would be less data to learn from for a domain shift [in CIFAR10/100-to-CIFAR10/100C].
> > >
> > > On the mentioned CIFAR->CIFAR-C task in the CoTTA paper, each corruption has 10,000 images which amounts to 156 adaptation steps with our current hyperparams. RDumb reaches on average 84.6% of its performance gain on each holdout noise already after 156 steps (Figure 5a, paper). Therefore, we think that our existing experiment already answers your original question of adaptation behavior with limited data per domain shift. However, please let us know if we missed something / please clarify what you meant originally!
> > >
> > > That being said, we still ran the experiments to address your remaining point:
> > >
> > > > Experimental results on these benchmarks [CIFAR10/100-to-CIFAR10/100C] would make the case stronger for the proposed approach.
> > >
> > > We can now share full results for RDumb, CoTTA, BN and a pre-trained net on C10-C and C100-C for the setting you outline. We expand over the CoTTA paper by running 10 permutations to be able to test for statistical significance and report error bars, as suggested by Reviewer Gw5H. Note that CoTTA uses augmentations that resemble the ones in CIFAR-C (e.g. brightness, Gaussian Noise, contrast, blurring, etc.) which we removed to facilitate a meaningful comparison to RDumb.
> > >
> > > RDumb significantly outperforms CoTTA in this setting on both CIFAR10 and CIFAR100, either when using the parameters reported in the paper, or when being re-tuned using the same protocol as RDumb (Table 1, below). Note, CoTTA also takes more than 11x as long to run as RDumb on both C10-C and C100-C (Table 2, below).
> > >
> > > Overall, we acknowledge your encouragement to let us run this experiment --- it shows that RDumb outperforms CoTTA in a comparable setup also on CIFAR10 and CIFAR100, while being considerably faster.
> > >
> > > As a side-note, the optimal reset parameter found by our hyperparameter search is larger than the whole dataset, i.e., the model never resets on this run --- this is exactly the intended optimal behavior we anticipated in [our original response](https://openreview.net/forum?id=VfP6VTVsHc&noteId=hp6DGsSHib) (CIFAR-C does not result in a collapse asymptotically, cf. Figure 4, [rebuttal PDF](https://openreview.net/forum?id=VfP6VTVsHc&noteId=SZYBZnJCYQ)). Our theoretical model of collapse proposes a possible explanation for this behavior: the dataset might be to easy (20.43% or 35.37% error after correcting BN stats vs. 57.4--93.2% error in CCC-easy to -hard, cf. Table 1, paper). For sufficiently high signal-to-noise ratios, collapse can be avoided (cf. Figure 1b, [rebuttal PDF](https://openreview.net/forum?id=VfP6VTVsHc&noteId=SZYBZnJCYQ)).
> > >
> > > Please let us know what you think and whether this addresses your remaining concern. We are also happy to further discuss your original motivation for us to run this experiment.
> > >
> > > ---
> > >
> > > *Table 1: Performance comparison between CoTTA and RDumb on the CIFAR→CIFAR-C tasks. We report error rates in % (mean +/- empirical standard deviation) over n=10 permutations of noise sequences. Differences between models are statistically signficant (ANOVA and Tukey HSD for post-hoc testing, RDumb vs. CoTTA at p<0.0001 in all settings).*
> > >
> > > |                          | CIFAR10-C       | CIFAR100-C     |
> > > |--------------------------|---------------------|---------------------|
> > > | Baseline                 | 43.28 +/- 0.00      | 46.66 +/- 0.00      |
> > > | Batch Norm               | 20.43 +/- 0.00      | 35.37 +/- 0.00      |
> > > | CoTTA (reported params, Wang et al., 2022)  | 18.02 +/- 0.214     | 33.20 +/- 0.247     |
> > > | CoTTA (validated params, ours) | 18.31 +/- 0.308     |                     |
> > > | RDumb (validated params, ours) | 17.27 +/- 0.196 | 31.84 +/- 0.195 |
> > >
> > > ---
> > >
> > > *Table 2: Runtime comparison for a full run through the dataset (150k samples in total) for each method. RDumb is 14.3x faster than CoTTA on CIFAR10, and 11.9x faster than CoTTA on CIFAR100. Note that a WRN-28 is used on CIFAR10 and an AugMix ResNeXt is used on CIFAR100 in line with Wang et al. (2022).*
> > >
> > > |            | CIFAR10 | CIFAR100 |
> > > |------------|---------------|-----------------|
> > > | Baseline   | 60.9s         | 33.5s           |
> > > | Batch Norm | 80.6s         | 36.1s           |
> > > | CoTTA      | 1909.7s       | 856.2s          |
> > > | RDumb      | 133.5s        | 71.9s           |

---

> > > > ### Author Response · Authors · 2023-08-21
> > > > **Thank you for your response.**
> > > >
> > > > Dear Reviewer,
> > > >
> > > > Thank you for your responses.

---

### Official Review · Reviewer_BezD · 2023-07-07

**Soundness:** 2 fair
**Presentation:** 2 fair
**Contribution:** 2 fair
**Rating:** 3
**Confidence:** 4

**Summary:**

The paper newly introduces a new benchmark dataset for Continual Test-Time Adaptation (CTTA) named Continually Changing
Corruptions (CCC), and suggests a simple technique - repeatedly initialize the learned model weights during CTTA. CCC is composed of interpolated corruption data and its data scale varies at each timestep. They can control the difficulty of CCC data by changing the degree of interpolation of two different corruptions.

**Strengths:**

The paper suggests a new large-scale continual test-time adaptation benchmark dataset, considering corruption interpolation, scale variation, and repetition.

**Weaknesses:**

The suggested method is too naive and not attractive in view of knowledge transfer. Periodically resetting the trainable weights to the initial state indicates simply discarding obtained knowledge and adaptive representation regardless of their importance and relevancy. Even though this remedy outperforms baselines, there is a lack of quantitative and qualitative analysis of why it happens and of the reason RDumb behaves differently from baselines. Additionally, the decision of resetting iteration is heuristic.

The suggested idea is surprising but I strongly recommend more rigorous analyses and validation of the suggested method and CCC dataset since they are counterintuitive in some sense.

**Questions:**

Why does RDumb consistently outperform a pre-trained model during TTA? Shouldn't its performance reach the pre-trained model's performance when re-setting the model weights to the initially pre-trained ones?

**Limitations:**

Please see the weakness.

---

> ### Author Rebuttal · Authors · 2023-08-10
>
> Thanks for your review! We agree with your assessment that our results are quite surprising. We hope to clarify your concerns and address the remaining weaknesses below:
>
> > “Even though this remedy outperforms baselines”
>
> This sounds like a possible misunderstanding. While RDumb indeed outperforms baselines (pre-training, batchnorm), it also outperforms (or matches) *all published state-of-the art test time adaptation methods*, which includes CoTTA and EATA, while being conceptually simpler and easier to analyze. Due to its technical simplicity, our paper argues that RDumb should serve as a baseline in follow up work, and questions the way we assess TTA methods.
>
> > The suggested method is too naive
>
> Technical complexity by itself is not a criterion for acceptance at NeurIPS: the principle of Occam's razor remains pertinent. Our results suggest that an extremely simple method performs just as well or *better* than more complex ones.
>
> But furthermore, besides the RDumb method, there is a lot of technical depth in the construction of CCC: Unlike previous datasets which simply stack a few existing data points, we propose a well-designed setup which is orders of magnitude bigger than current evaluation setups, in number of noises (210 vs 15), number of severities (441 vs 5).
>
> It enables the benchmarking of continual methods for at least 10 times as long as current benchmarks, without repeating images (and longer, if required in the future). The complexity of the benchmark allows for evaluating methods on a controlled baseline accuracy, which wasn’t possible in previous work.
>
> With CCC, we pose a now challenging evaluation setting which highlights the shortcomings of current TTA methods. This evaluating setting is arguably the first true “continual test-time adaptation setting”, as previous benchmarks miss important hallmarks of continual adaptation (as we argue).
>
> This is a key contribution of this work, and its depth is easy to miss, as a lot of the work of constructing the benchmark is hidden in the supplementary material.
>
> We’re happy to discuss this point further in case we missed what you meant.
>
> > … not attractive in view of knowledge transfer. Periodically resetting the trainable weights to the initial state indicates simply discarding obtained knowledge and adaptive representation regardless of their importance and relevance.
>
> We respectfully disagree.
>
> The fact that such a method outperforms others suggests that *none* of the previously used methods are effective at knowledge transfer, and that is a very valuable insight (as you point out) that was missed in previous work. The title of our paper exactly reflects this line of thought.
>
> While other methods are allowed to benefit from many images seen, they are still not as effective as a method that resets itself every 1,000 steps.
>
> This is a very surprising result given claims in previous papers [1,2]. Don’t you agree? Happy to discuss further.
>
> > lack of quantitative and qualitative analysis of why it happens
>
> This is a fair point. We plan to add the following analysis that will further strengthen the paper:
>
> We constructed a simple 2d Gaussian binary classification example, a domain shift which slightly rotates the data and adds Gaussian noise, and a model which consists of a batch norm layer followed by a logistic regression. When running entropy minimization on the adaptation layer of the batch norm, simple cases emerge where the loss does or does not result in collapse, mainly depending on the relation of signal and noise variances and directions (Fig. 1-2, Rebuttal PDF).
>
> The toy example furthermore predicts that the adapted parameters of a model should grow on the long run and indeed we were able to find exactly this effect when running ETA on a ResNet50 on CCC-Medium (Fig. 3, rebuttal PDF), suggesting that our minimal setup successfully reproduces the relevant aspects of the large scale case. However, the weight explosion becomes apparent only after the collapse happens, hence weight regularization is not enough to avoid the collapse (see Fig 3b).
>
> Thank you for raising this issue, we think that this analysis adds to the paper and we will include it in the camera ready version. We would be happy to hear your thoughts on this experiment.
>
> > the decision of resetting iteration is heuristic.
>
> This is not the case, please see Section 6. The reset interval is obtained by cross-validation on the holdout noises, which to our knowledge is the known best practice in the field. However, we are happy to re-evaluate with another selection strategy, if you have suggestions (but doubt it would change the core message of our paper).
>
> As an aside, we would argue that a resetting interval is a more interpretable hyperparameter than a resetting fraction (in CoTTA) or regularizer weights (in EATA).
>
> We’d be happy to discuss this point further with you.
>
> > Q: Why does RDumb consistently outperform a pre-trained model during TTA? Shouldn't its performance reach the pre-trained model's performance when re-setting the model weights to the initially pre-trained ones?
>
> You’re right that the pretrained model and RDumb have similar accuracies everytime RDumb is reset. In Figure 2, each point is an average of 781 batches (approx. 50k images, i.e. the size of the ImageNet validation set). We do this because otherwise there would be too many points on the graph. RDumb is able to learn quickly and therefore surpasses the pretrained model within those 781 batches, as can also be seen in Figure 5a. We will make this clearer in the next draft, thank you for bringing this to our attention.
>
> We hope this addresses your concerns, and would be happy to discuss further.
>
> [1] Wang, Qin, et al. "Continual test-time domain adaptation." Proceedings of the IEEE/CVF Conference on Computer Vision and Pattern Recognition. 2022.
>
> [2] Niu, Shuaicheng, et al. "Efficient test-time model adaptation without forgetting." International conference on machine learning. PMLR, 2022.

---

### Official Review · Reviewer_h5ar · 2023-07-24

**Soundness:** 3 good
**Presentation:** 4 excellent
**Contribution:** 3 good
**Rating:** 7
**Confidence:** 3

**Summary:**

The paper proposes a new benchmark (dubbed CCC) for test-time adaptation, which generalizes previous corrupted imagenet benchmarks. Specifically, the CCC gradually draws a sequence of corruptions (e.g. gaussian noise or motion blur), and gradually interpolates between two consecutive corruptions, creating a stream without hard boundaries, akin to many realistic settings. By considering long sequences of corruption pairs, CCC yields a stream longer than previous benchmarks, which sheds new light on how previous methods fair in such settings. Indeed, it is shown that given a long enough stream, methods collapse to a performance worse than a non-adapted model. The authors then propose a new baseline, Rdumb, which resets to the default parameters of the pretrained model at regular, fixed intervals. Over both CCC and previous benchmarks, it is shown that Rdumb, despite its simplicity, works well.

**Strengths:**

1. The proposed benchmark is well-designed. I liked the use of calibration with a pretrained model to generate different stream difficulties, and the fact that this overall is stream generator can is flexible and can accommodate smooth transitions.
2. The proposed baseline, Rdumb, is extremely simple and should be considered as a standard baseline for TTA.
3. Overall the paper is well written and easy to follow.

**Weaknesses:**

1. The idea of resetting the model to its original pretrained state in order to combat accumulating degradation in TTA is not new (as correctly stated in the paper). Indeed, it was shown in [45] that resetting after each task boundary helps the network. For a fair comparison with COTTA, how would such method perform if the probability of resetting the weights was determined by a similar cross-validation technique as used in Section 6 ?
2. COTTA also has some experiments with gradually changing corruptions (see Table 3). it would be good for this to be mentioned in the paper.

minor comments :
1. regarding figure 2, I would use the color red as the most severe corruptions, and yellow as the least severe.
2. Optimal resetting interval : I think it should be Rdumb, and not Gdumb (lines 189-190)

**Questions:**

1. My understanding of TTA's use-cases is mostly for online settings; the deployed model receives data for which to output predictions, and this data may encounter distribution shifts. In such a setting, how would one determine the frequency of when to reset the model ?

---

> ### Author Rebuttal · Authors · 2023-08-10
>
> Thanks for your review. We’re happy you found our dataset interesting, and our paper easy to read. We want to address the weaknesses and questions below.
>
> > For a fair comparison with COTTA, how would such method perform if the probability of resetting the weights was determined by a similar cross-validation technique as used in Section 6?
>
> This is a good suggestion. To recap, CoTTA has a standard parameter of p=0.001 (on ImageNet), which we used in the experiments so far. We now applied our full evaluation protocol to CoTTA, and checked different thresholds on the holdout set:
>
> | resetting param (p) | 0.001 (default) | 0.005 | 0.01 | 0.05 | 0.1 | RDumb |
> |--------------------|----------------|-------|------|------|-----|-------|
> | CIN-C Holdout Avg. Acc    | 35.73          | 36.77 | 38.04| 37.86| 37.41| 46.7  |
>
>
> We find an optimal parameter of p=0.01 which we apply to our test set:
>
> |              	| CCC-Easy* | CCC-Medium | CCC-Hard |
> |------------------|-------------------|------------|----------|
> | Pre-Trained  	| 34.04         	| 17.3   	| 1.5  	|
> | Cotta (p=0.001)  | 14.9          	| 7.7    	| 1.1  	|
> | Cotta (p=0.01)   | 27.8          	| 15.6   	| 1.1  	|
> | RDumb        	| 49.15         	| 38.9   	| 9.6  	|
>
> *\*Note: All runs except for one are finished on CCC-Easy, therefore we take out the unfinished run from the metrics of all models to have a fair comparison. We will of course update this table with the finished run as soon as possible.*
>
> While the results seem to improve slightly, note that CoTTA is still worse than a pretrained net on all benchmark datasets. We suggest to include this analysis in the supplementary material, and use the published hyperparameters (which is also present in the CoTTA codebase), as the message remains consistent.
>
> Does this clarify your concern? Thanks again for this good suggestion.
>
> > COTTA also has some experiments with gradually changing corruptions (see Table 3). it would be good for this to be mentioned in the paper.
>
> Yes you are right, thanks for pointing this out, we now call this out in the discussion of previous work.
>
> For completeness, however, we would like to point out three important differences and contributions CCC makes over this setting:
>
> First, the setting used in the COTTA paper is way easier than CCC and fails to show a collapse to chance level. We ran an experiment on the concatenated CIFAR10-C dataset for more than 100 million images (vs. 150k images in the CoTTA paper) with no sign of collapse (Figure 4, Rebuttal PDF). We also run a similar experiments for 25 million images on the gradual CIFAR10 task (Figure 5, Rebuttal PDF). While the fluctuations in performance increase, the setup still fails to produce the collapsing behavior to chance level we observe at ImageNet-scale with CCC. This highlights the difference in difficulty to our “real world sized” CCC dataset.
>
> Second, “gradual” here means a transition from clean to corrupted images. CCC corruptions have a much larger combinatorial space due to the simultaneous application of multiple corruptions.
>
> Third, the C10-C tasks are less-well controlled, which is similar to the issues we discuss for concatenated ImageNet-C. You can see this in Figure 6 in the rebuttal PDF, where performance fluctuates greatly between noises. This makes the dataset not suitable for controlled evaluation of models in a scientific setting.
>
> The point you raise will help to position CCC even better within the existing literature, thanks for the suggestion.
>
> > My understanding of TTA's use-cases is mostly for online settings; the deployed model receives data for which to output predictions, and this data may encounter distribution shifts. In such a setting, how would one determine the frequency of when to reset the model ?
>
> Resetting is not conceptually different from any other hyperparameter in TTA models, your concern would hence also apply to the hyperparameters in CoTTA or EATA.
> However, in contrast to previous work, the resetting parameter has the advantage of being easily interpretable. As it tends to 0, the model gets more conservative, and approaches the performance of batch norm adaptation. As it tends to infinity (no resetting), the model approaches the performance of a (collapsing) TTA algorithm.
> We show (cf. Table 2) that the model performance on CCC is fairly robust to the exact setting. In real-world scenarios, the parameter could be easily set by cross-validation (like any other hyperparameter in TTA) on a suitable development set (which are now very easy to generate for different difficulty levels and noise settings thanks to CCC).
>
> We hope this addresses your concern, and we are happy to answer any questions that might still remain.

---

> > ### Comment · Reviewer_h5ar · 2023-08-12
> > **Answer to Rebuttal**
> >
> > Thank you for the rebuttal.
> >
> > After reading the other reviewers' comments and your rebuttal,
> >
> > - I agree that lacking Technical complexity is not a negative attribute in itself; simple methods that work well should be prioritized.
> > - I agree that an analysis as to *why* existing methods suffer from such performance degradation would a good addition to the paper. I thank the authors for the additional experiments, and making some progress towards answering this.
> >
> > While my expertise is not directly in continual test time adaptation, I do believe that well executed papers which revisit the current state of a field of research (and its evaluation protocols), as well as propose simple methods which perform well, can be of value to the research community. I believe that my previous score and confidence reflect my overall assessment of the paper, and will therefore leave as-is.
> >
> > Thank you.

---

> > > ### Author Response · Authors · 2023-08-21
> > > **Thank you for your response**
> > >
> > > Dear Reviewer,
> > >
> > > Thank you for your responses.

---

### Author Rebuttal · Authors · 2023-08-10

We thank the reviewers for their constructive feedback. We are glad to hear that reviewers found our paper easy to follow, our proposed dataset interesting, and our method to be simple and effective.

We commented on all reported weaknesses and addressed the reviewers' questions in the individual responses below.

Concerning new results to be added to the paper, we report new results for [additional hyperparameter tuning on CoTTA](https://openreview.net/forum?id=VfP6VTVsHc&noteId=zMGLcCwMDu), an [updated Table 1](https://openreview.net/forum?id=VfP6VTVsHc&noteId=rZV3x9TDSH) with error bars, and extensive new analysis of the collapsing behavior, for which we attach additional figures in the PDF here.

We hope that our replies clarify all reviewer concerns, and are happy to engage in further discussion and experiments.

---

### Author Response · Authors · 2023-08-21
**Discussion Phase Summary**

Dear AC,


Overall, we enjoyed a constructive discussion phase and the reviewers found our work meaningful, thorough, easy to understand, and found our contribution to be significant. The current scores are: 7,7,5,3 (and we are waiting for replies from the latter two). During the discussion phase, we did the following:


- **Added analysis on collapse in a toy setting**: Reviewers BezD and h5ar asked for additional analysis/discussion on the causes for collapse. We now added a theoretical analysis that addresses both settings of model collapse and stable behavior and links this to the signal-to-noise ratio in the data. We confirm that the learning dynamics match between small scale (simulated 2D data) and large scale (CCC scale) experiments. The analysis sheds light on the causes for collapse during adaptation, indicating why RDumb (full model resetting), CoTTA (gradual model resetting) and EATA (regularization-based) outperform standard entropy minimization techniques.
- **CIFAR10-C/CIFAR100-C experiments**: Reviewers h5ar and xRL5 asked for additional results on CIFAR scale. While we presented arguments of important shortcomings of CIFAR vs. our proposed CCC benchmark, we also ran these results. We now show that RDumb also significantly outperforms CoTTA in a comparable experimental setup on CIFAR, further corroborating our larger-scale experimental results. While the reviewer did not reply back yet, we hope that these new results address any remaining doubts about RDumb’s effectiveness.
- **Additional CoTTA experiments**: Reviewer h5ar asked us to run CoTTA with more hyperparameters. We show that CoTTA still fails after an extensive hyperparameter search.

The only reviewer, BezD, that currently leans towards rejection didn’t participate at all in the discussion. Their main reason for rejection was that our method was allegedly “too simple” - but that was our intention from the get-go: showing that an extremely simple method performs just as well or *better* than the more complex SOTA ones.. We believe that the importance of Occam’s Razor principle is still pertinent. Reviewers h5ar (“simple methods that work well should be prioritized”) , xRL5 (“A simple periodic reset mechanism with weighted entropy … outperforms the SOTA CTTA approaches” [under strengths]), and Gw5H (“The baseline is simple and satisfies all the TTA assumptions, and should be considered as a baseline in future works in this area” [under strengths]) agreed with us that the simplicity of our method is an *advantage*.


Thank you for taking this into consideration.

---

### Decision · Program_Chairs · 2023-09-21

**Decision:**

Accept (poster)

**Comment:**

This work on test-time adaptation contributes a new benchmark (CCC) for continual adaptation, which mixes corruptions in a continuously changing fashion, and a simple method (RDumb), which periodically resets the model and can serve as a baseline for more sophisticated methods. Four reviewers with expertise in adaptation, continual learning, and personalization vote for acceptance (h5ar : 7, Gw5H: 7, xRL5: 5) and rejection (BezD: 3). The authors provide a rebuttal with experiments, and all reviewers respond to the authors except for BezD, as noted by the AC and mentioned by an author comment. The main argument for acceptance is that the RDumb method and CCC benchmark clearly question how well continual adaptation methods continue to adapt, and the question is justified by the thorough experiments. The main argument for rejection are lack of analysis or insight for why continual adaptation degrades (xRL5, BezD), lack of novelty w.r.t. continual adaptation (h5ar, xRL5), and insufficient development of how to reset (BezD, h5ar). The rebuttal addresses each of these negatives. While further analysis—of the dataset or adaptation degradation—and more sophisticated resetting would improve this work, the AC sides with acceptance, because the empirical content is already informative (even surprising!) and rigorous. As such this work can guide the next steps for continual adaptation and gauge progress in this direction.

Note: reviewer BezD engaged in AC-reviewer discussion to confirm the vote for rejection. They emphasize 1. the limitation of resetting as a method, 2. lack of analysis that explains how resetting succeeds while existing methods fail, and 3. the need to further validate the CCC benchmark in its ordering, scaling, and repeating of corruptions. Re: 1., the limitation of resetting is real, in particular the inability to forward transfer knowledge (say for cyclic shifts like seasons or repeated shifts like weather), the experiments on resetting show it to be a valid baseline. Re: 2. and 3., more analysis is always better (and specifically expanding the rebuttal experiments to cover ETA/EATA would show more generality), but the AC and the majority of reviewers see the current experiments as sufficient. The AC advises the authors to reflect on point 1. and consider addressing this in the discussion or conclusion, and the AC thanks BezD for their reasoned and specific points for further improvement.

The AC encourages the authors to incorporate the rebuttal material into the supplement for completeness, in particular the analysis of a toy example and the CIFAR-10/100-C results, as certain readers will find these informative.